# Lineage-specific intersection of endothelin and GDNF signaling in enteric nervous system development

**Denise M Poltavski†, Alexander T Cunha†, Jaime Tan, Henry M Sucov, Takako Makita***

Department of Regenerative Medicine and Cell Biology, Medical University of South Carolina, Charleston, United States

## eLife Assessment

This study provides **valuable** insights into our understanding of the development of the enteric nervous system. The authors use genetically engineered mice to study the behavior of stem cells in organizing the enteric nervous system and the secreted signals that regulate these cells. The study rests on a degree of **incomplete** evidence since the characterization of some of the mouse resources is not complete in the current version.

***For correspondence:**
makita@musc.edu

†These authors contributed equally to this work

**Competing interest:** The authors declare that no competing interests exist.

**Abstract** Two major ligand-receptor signaling axes – endothelin Edn3 and its receptor Ednrb, and glial-derived neurotrophic factor (GDNF) and its receptor Ret – are required for migration of enteric nervous system (ENS) progenitors to the hindgut. Mutations in either component cause colonic aganglionosis, also called Hirschsprung disease. Here, we have used Wnt1Cre and Pax2Cre in mice to show that these driver lines label distinct ENS lineages during progenitor migration and in their terminal hindgut fates. Both Cre lines result in Hirschsprung disease when combined with conditional *Ednrb* or conditional *Ret* alleles. In vitro explant assays and analysis of lineage-labeled mutant embryos show that GDNF but not Edn3 is a migration cue for cells of both lineages. Instead, Edn3-Ednrb function is required in both for GDNF responsiveness albeit in different ways: by expanding the Ret+ population in the Pax2Cre lineage, and by supporting Ret function in Wnt1Cre-derived cells. Our results demonstrate that two distinct lineages of progenitors give rise to the ENS, and that these divergently utilize endothelin signaling to support migration to the hindgut.

## Introduction

The enteric nervous system (ENS) is the largest division of the peripheral nervous system and harbors a complex circuitry. Approximately a half billion enteric neurons in human (and over a million in mice) distribute along the entire axis of the gastrointestinal tract to regulate a variety of physiological functions (**Furness, 2006**). Colonic peristalsis that promotes defecation is controlled by several subtypes of enteric neurons that reside in the myenteric plexus that is located between the longitudinal and circular muscles of the gut. Colonic myenteric mechanosensory neurons are multiaxonal (Dogiel type II) neurons of which one axonal branch extends to the gut mucosa and synapses with mechanosensory (enterochromaffin) cells, while the opposite branches synapse on motor neurons or interneurons in neighboring myenteric ganglia. When mucosal enterochromaffin cells are stimulated by the presence of feces, mechanosensory endings are activated and then transmit signals through interneurons to activate or inhibit motor neurons, which generate contraction or relaxation of colonic circular muscle,

resulting in peristalsis. This neural circuitry enables the ENS to sustain autonomous gut motility in the absence of central nervous system input.

The entirety of the ENS has been thought to arise exclusively from migratory neural crest cells. This conclusion was based primarily on avian intraspecies cell transplantation (*Le Douarin, 1982*) and dye labeling studies (*Serbedzija et al., 1989*), and subsequently supported by genetic labeling studies of the neural crest cell lineage in mice (*Yamauchi et al., 1999*; *Kapur, 2000*; *Yu et al., 2024*). Enteric neural crest cells arise from the vagal region of the hindbrain and cervical spinal cord, migrate first to the foregut, and then continue to migrate along and colonize the entire length of the gut. Once in the gut, these organize into intrinsic ganglia, including the colonic myenteric plexus that has primary control of gut motility. A distinct sacral neural crest cell population colonizes the hindgut, migrating from the distal end of hindgut in a cranial direction (*Anderson et al., 2000*). The majority of sacral neural crest cells constitute the pelvic ganglia that are located on the rectal wall and project rostrally along the colon to supply sympathetic and parasympathetic control of gut physiology, but not including colonic peristalsis (*Keast, 1995*). Neural crest clearly contributes substantially to the ENS, and while past studies could not formally exclude a contribution from other sources, there has also been no experimental observation that would invoke the involvement of any other developmental origin.

Defective caudal migration of enteric progenitors results in aganglionosis in the terminal bowel. This has no consequence in fetal life while gut motility is suppressed, but after birth, when fecal progression through the hindgut requires a functional ENS circuitry, absence of adequate ENS function leads to congenital gut motility disorders, characterized by the inability to defecate and involuntary fecal retention. In such cases, the hindgut expands to accommodate the accumulating material, accounting for one name of this condition as congenital megacolon. A synonymous term for this pathology is Hirschsprung disease.

Hirschsprung disease is an inherited or sporadic condition in humans, and is associated with mutations in genes encoding two pairs of extracellular ligand/receptor signaling molecules: the endothelin ligand EDN3 and its G-protein-coupled receptor (GPCR) EDNRB, and the receptor tyrosine kinase RET and its ligand GDNF (*Parisi and Kapur, 2000*). Mice that carry mutations in these genes lack intrinsic ganglia in the distal colon and therefore phenocopy the human condition (*Baynash et al., 1994*; *Hosoda et al., 1994*; *Schuchardt et al., 1994*; *Durbec et al., 1996*; *Sánchez et al., 1996*). Hirschsprung disease can present alone or as a component of Waardenburg-Shah syndrome that includes deafness and pigmentation abnormalities, reflecting the contributing role of these signaling pathways in other tissues, but colonic aganglionosis is the defining condition of this syndrome. Hirschsprung disease is also a prototypical example of a large class of congenital syndromes and adult cancers collectively called neurocristopathies that involve neural crest cells as a common etiologic factor (*Mueller and Goldstein, 2022*).

Placodes are bilateral condensations of the cranial surface ectoderm that generate migratory neurons and other cell types in a manner that is similar to neural crest (*Baker and Bronner-Fraser, 2001*). Placodes and neural crest jointly contribute to many cranial ganglia, including the Xth cranial (vagus/nodose) ganglia. While neural crest gives rise to a variety of neuronal subtypes and glia, neurogenic placodes are only known to give rise to sensory neurons (*D'Amico-Martel and Noden, 1983*). The contribution of the placode cell lineage has only been recognized in the head but never before associated with the ENS.

In a recent analysis (*Tan et al., 2023*), we used Pax2Cre to label the placode lineage and Wnt1Cre to label the neural crest cell lineage in mice to demonstrate their contributions to specific components of the auditory system. As documented in past studies (*D'Amico-Martel and Noden, 1983*, *Barald and Kelley, 2004*; *Ladher et al., 2010*) and as reiterated in our study with Pax2Cre and Wnt1Cre as lineage markers (*Tan et al., 2023*), the placode and neural crest cell lineages have distinct and independent fates in the inner ear. Auditory functionality requires endothelin (specifically Edn3) signaling through the endothelin receptor Ednrb in both lineages, although these requirements are separable and independent and have different consequences, such that perturbation of either results in congenital deafness.

In this study, we have used Pax2Cre and Wnt1Cre to address the contribution of these cell lineages to the ENS. We demonstrate that CGRP-immunoreactive colonic myenteric mechanosensory neurons are uniquely labeled by Pax2Cre, whereas NOS$^+$ inhibitory motor neurons are uniquely derived from

the Wnt1Cre lineage. Importantly, Edn3-Ednrb, a signaling axis previously defined to be important for enteric neural crest cell migration, also controls the migration of Pax2Cre-labeled enteric progenitor cells, such that conditional *Ednrb* mutation driven by Pax2Cre or Wnt1Cre both result in compromised gut motility. We further show that GDNF-Ret signaling is required in both lineages to support ENS progenitor cell migration, albeit in a manner that intersects with Edn3-Ednrb signaling differently in Wnt1Cre-derived and Pax2Cre-derived cells. Our findings implicate a dual lineage origin for the ENS with mechanistically distinct features, and illuminate the consequences of Edn3 and GDNF signaling that account for Hirschsprung disease when either is compromised.

## Results

### Impaired colonic peristalsis in the absence of Pax2Cre-derived ENS cells in the distal colon

In our recent study addressing the placode lineage contribution to auditory development in mouse models for Waardenburg-Shah syndrome (*Tan et al., 2023*), we generated neural crest-specific (using Wnt1Cre) and placode-specific (using Pax2Cre) conditional *Ednrb* homozygous mutant mice (designated as *Wnt1Cre/Ednrb* and *Pax2Cre/Ednrb* respectively). In the outbred ICR background of our colony, we found that neither *Wnt1Cre/Ednrb* mutant nor *Pax2Cre/Ednrb* mutant mice lived beyond 3 weeks of age (*Tan et al., 2023*). The lethality of *Wnt1Cre/Ednrb* mutants around postnatal day P21 was predicted because of the well-established role of Edn3-Ednrb signaling in enteric neural crest migration and colonization (*Baynash et al., 1994*; *Hosoda et al., 1994*). However, the occurrence (and full penetrance) of lethality around this same time in *Pax2Cre/Ednrb* mutants was unexpected, as there was no prior implication that placode cells contribute to the ENS or do so in an endothelin signaling-dependent manner. *Pax2Cre/Ednrb* mutant mice exhibited growth retardation and reduced weight gain comparable to global *Ednrb* and conditional *Wnt1Cre/Ednrb* mutants in the second postnatal week (*Figure 1a and b*). Dissection and inspection of the gastrointestinal tract revealed that *Pax2Cre/Ednrb* mutants suffer from colonic distention and constriction similar to global *Ednrb* and *Wnt1Cre/Ednrb* mutant mice (*Figure 1d*), associated with near complete abrogation of defecation (*Figure 1c*). These observations indicate that *Ednrb* deficiency in the Pax2Cre lineage impairs colonic peristalsis.

To visualize the contribution of Pax2Cre-derived cells to the ENS and its dependence on endothelin signaling, we crossed the *Rosa26^lacZ* reporter into the *Pax2Cre/Ednrb* background and visualized Pax2Cre-labled cells during ENS progenitor cell migration and colonization. We conducted a parallel analysis for the Wnt1Cre lineage. Fate mapping with *Pax2Cre/Rosa26^lacZ* in normal embryos showed labeled cells reaching the midgut loop at E11.5, reaching the mid-hindgut junction at E12.5, and nearly completing the colonization of the hindgut by E13.5 (*Figure 1—figure supplement 1f–h*, *Figure 1e*). A comparable pattern was observed with Wnt1Cre-derived migratory cells (*Figure 1—figure supplement 1b–d*, *Figure 1g*), and this temporal pattern has been previously observed using a variety of nonlineage-dependent markers of enteric progenitors (*Lake and Heuckeroth, 2013*). *Ednrb*-deficient Pax2Cre lineage-labeled cells (*Figure 1f*) and *Ednrb*-deficient Wnt1Cre-derived cells (*Figure 1h*) both failed to advance beyond the cecum by E12.5, and this ultimately resulted in a lack of cells from both lineages in the distal colon (hindgut-derived) of postnatal mice (*Figure 1j and m*). *Pax2Cre/Rosa26^lacZ* and *Wnt1Cre/Rosa26^lacZ* mice homozygous for a global *Edn3* mutant allele also exhibited an absence of lineage-labeled cells in the distal colon (*Figure 1k and n*), demonstrating that migration failure is for lack of endothelin signaling and not some artifact of the conditional *Ednrb* allele. *Edn3* ligand was expressed normally in the developing gut mesenchyme of both *Ednrb* conditional mutant backgrounds (*Figure 1f and h*). These results indicate that the Wnt1Cre lineage and the Pax2Cre lineage both populate the hindgut, and that both lineages utilize Edn3-Ednrb signaling for migration and colonization of the hindgut.

### Wnt1Cre and Pax2Cre define distinct ENS cell lineages

In the inner ear, the distinct and nonoverlapping fates of the Pax2Cre and Wnt1Cre cell lineages correspond precisely to placodal and neural crest fates as defined by prior studies (*Tan et al., 2023*). In addition to all known placode progeny (e.g. otic vesicle; *Figure 1—figure supplement 1i*) and its later derivatives in the inner ear and cranial ganglia (*Figure 1—figure supplement 1j–l*), Pax2Cre is

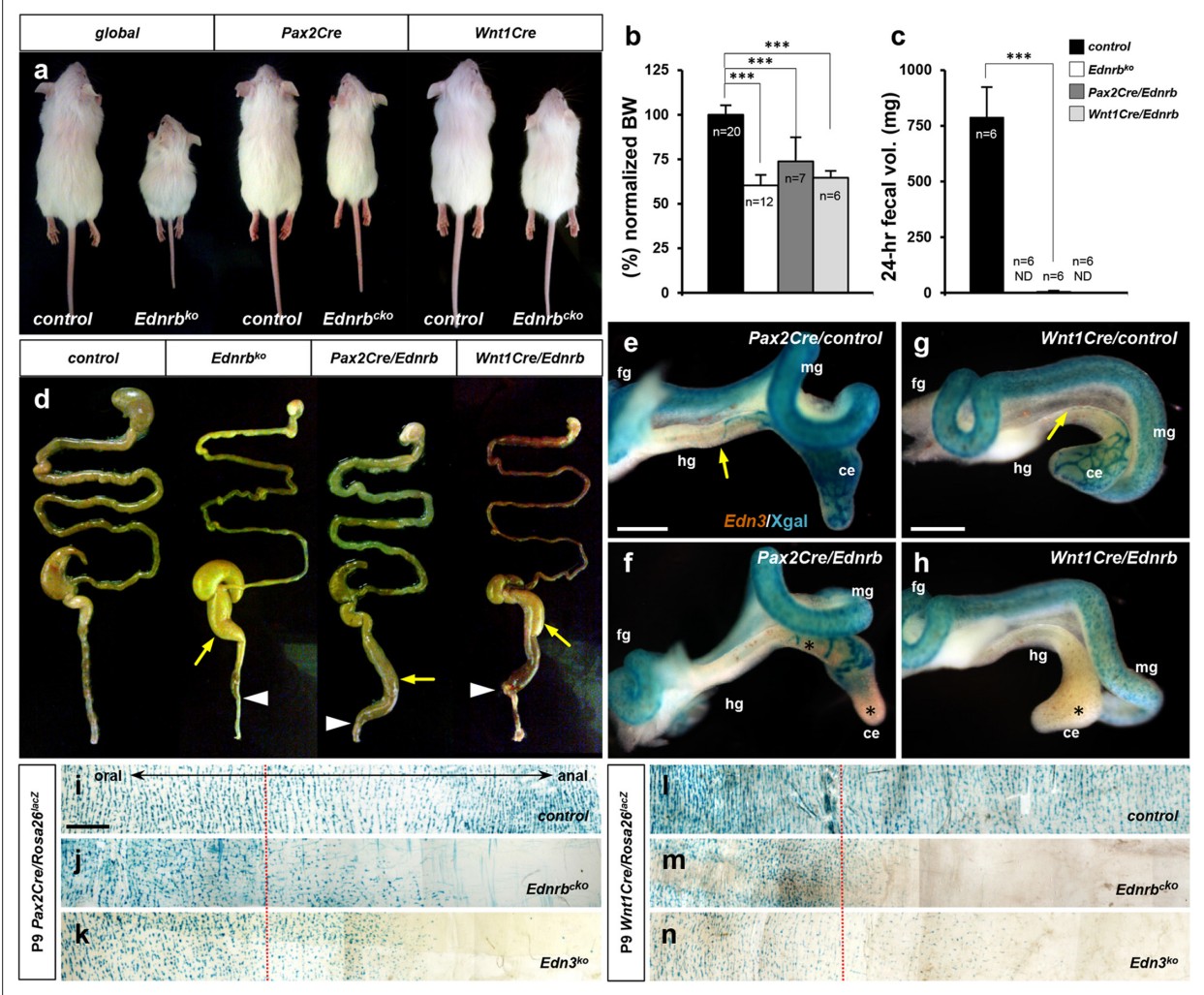

**Figure 1.** *Pax2Cre/Ednrb* mutants mice exhibit impaired colonic peristalsis. (**a**) External appearances of 2.5-week-old global *Ednrb* (left), *Pax2Cre/Ednrb* (middle), and *Wnt1Cre/Ednrb* (right) mutant mice compared to their littermate controls. (**b, c**) Compiled representation of normalized body weights (**b**) and 24 hr fecal elimination (**c**) from P18 mice of indicated genotypes. Error bars; mean ± s.d. ND, not detected. p-Value: ***p<0.001. (**d**) External appearances of P18 guts from the indicated genotypes. Arrows indicate distention of colon. Arrowheads point to constriction in the distal colon. (**e–h**) Wholemount preparations of E12.5 guts isolated from *Pax2Cre/Ednrb/Rosa26^{lacZ}* (**f**), *Wnt1Cre/Ednrb/Rosa26^{lacZ}* (**h**) mutant embryos and their littermate controls (e and g, respectively) stained with Xgal and labeled by in situ hybridization for *Edn3* expression (brown). Yellow arrows denote *Pax2Cre*-labeled (**e**) or *Wnt1Cre*-labeled (**g**) Xgal⁺ cells reaching the proximal-distal colon junction in control embryos, whereas *Ednrb*-deficient *Pax2Cre*-labeled cells (**f**) or *Wnt1Cre*-labeled cells (**h**) are deficient in caudal migration (asterisks) at the time when the *Edn3* gene is normally expressed. (**i–n**) Wholemount preparations of P9 distal colons isolated from *Pax2Cre/Ednrb* (**j**), *Edn3* null (**k**), and control (**i**) mice, and *Wnt1Cre/Ednrb* (**m**), global *Edn3* null (**n**), and control (**l**) mice stained with Xgal to visualize lineage-labeled cells. Dotted lines indicate the junction between supply by the superior mesenteric and inferior mesenteric arteries, the latter being the vascular source for the distal colon. Abbreviations: ce, cecum; fg, foregut; hg, hindgut; mg, midgut. Scale bars, 500 µm (**e–h**), 1000 µm (**i–n**).

The online version of this article includes the following source data and figure supplement(s) for figure 1:

**Source data 1.** Raw data for panel b (body weight) and panel c (fecal volume).

**Figure supplement 1.** Pax2Cre-derived cells contribute to the enteric nervous system (ENS).

**Figure supplement 2.** Pax2Cre and Wnt1Cre label distinct early migratory cell populations.

**Figure supplement 3.** Distinct lineage origin of Pax2Cre-labeled and Wnt1Cre-labeled enteric nervous system (ENS) progenitor cells.

active in several other cell types. Pax2Cre activity was detected in the ventral neural tube (*Figure 1— figure supplement 1i*), which does not overlap the Wnt1Cre-active (dorsal) domain. Additional Pax2Cre active cell types include limb mesenchyme (*Figure 1—figure supplement 1e*) and intermediate mesoderm and its derivatives (the mesonephric ducts and then later also in the ureteric bud)

(*Figure 1—figure supplement 1e and o*), although these sources are not migratory or neurogenic. Pax2Cre activity was not detected in several known postotic cranial/cardiac and trunk neural crest derivatives including dorsal root ganglia and sympathetic ganglia (*Figure 1—figure supplement 1m and n*) and mesenchyme of the cardiac outflow tract (*Figure 1—figure supplement 2c*). Despite this specificity, a trivial explanation for the occurrence of Hirschsprung disease phenotypes in both *Wnt1Cre/Ednrb* and *Pax2Cre/Ednrb* mutants could be if both Cre drivers recombine in the same cell populations of the ENS. As it is not possible to combine the two Cre alleles together to visualize their separate (or potentially overlapping) recombination domains, we addressed this by using molecular markers at the initial onset of their embryonic migration and in terms of the terminal (postnatal) fates of each lineage. The results, described below, confirm that Wnt1Cre and Pax2Cre define distinct ENS lineages.

First, we examined the expression domain of Pax2 protein with respect to the recombination domains of Pax2Cre and Wnt1Cre at the very early enteric progenitor migratory stage (E9.5–10.0). Prominent endogenous Pax2 was detected in the lens and otic vesicle (both known placode derivatives), in intermediate mesoderm (future kidneys), and germane to this report also in the postotic pharyngeal arches that abut vagal neural crest (*Figure 1—figure supplement 2a and b*). Pax2 protein overlapped Pax2Cre-labeled pharyngeal ectoderm and Pax2Cre-labeled mesenchymal cells just at the stage of delamination from the ectoderm, although Pax2 expression was downregulated with migration and therefore not detected in most Pax2Cre lineage-labeled mesenchymal cells in the postotic domain (*Figure 1—figure supplement 2c and d*). In contrast, Pax2 expression was excluded from the territory of pharyngeal neural crest marked by the Wnt1Cre lineage label (*Figure 1—figure supplement 2e and f*).

We also examined expression of neural crest markers AP2 (*Mitchell et al., 1991*) and Sox10 (*Southard-Smith et al., 1998*). AP2 and Sox10 were both detected within postotic migratory pharyngeal cells (*Figure 1—figure supplement 3a and b*). We observed that the entirety of Wnt1Cre-labeled cells expressed AP2 (*Figure 1—figure supplement 3e and f*). Sox10 expression was associated with a subset of the Wnt1Cre-labeled cell population, and thus with a subset of AP2-expressing arch mesenchymal cells (*Figure 1—figure supplement 3e and f*). In contrast, although there was overlap of AP2 expression with Pax2Cre in the ectoderm, neither AP2 nor Sox10 were expressed in any migratory Pax2Cre lineage-labeled cells (*Figure 1—figure supplement 3g and h*).

We then examined the enteric progenitor marker Ret during the early stage of enteric progenitor migration in the gut (E10.0–10.5) in normal *Pax2Cre/Rosa26$^{nT-nG}$* and *Wnt1Cre/Rosa26$^{nT-nG}$* embryos by two-color immunohistochemistry for nuclear-GFP and Ret (*Figure 2*). We quantified the number of nGFP-positive, Ret-positive, and nGFP and Ret-double positive cells; cells that did not express either the lineage marker or Ret were excluded from measurement as these are indistinguishable from all other cell types that are present in the gut. In *Pax2Cre/Rosa26$^{nT-nG}$* embryos (six embryos from different litters, average 249.7±15.7 counted cells per embryo), only 11.8±1.2% of the counted cells were Pax2Cre-derived (*Figure 2v*, black and gray bars); the other 88% (*Figure 2v*, white bar) are presumed to be Wnt1Cre-derived Ret$^+$ cells (which was directly measured at 86% as described below). Ret was expressed in 14.4±1.7% of the Pax2Cre lineage-labeled cells, but only 1.6±0.2% of the total counted cells. We performed the same analysis for the Wnt1Cre lineage (six embryos; 178.8±11.4 counted cells per embryo). Here, almost all (97.8±0.2%) counted cells were lineage labeled, and of these, 86.2±1.5% expressed Ret. Only 2.2±0.2% of counted cells in *Wnt1Cre/Rosa26$^{nT-nG}$* embryos expressed Ret but were lineage-negative; these are presumably Pax2Cre-derived, which was directly measured at 1.6% as described above. Consequently, 97.5% of the Ret-expressing cells in *Wnt1Cre/Rosa26$^{nT-nG}$* embryos were of this lineage. Approximately 10–15% of lineage-labeled cells in both analyses did not express Ret (*Figure 2v*, gray bars). These results indicate that Ret-expressing cells in the E10.5 midgut are overwhelmingly derived from the Wnt1Cre lineage. The majority of Wnt1Cre-derived cells at E10.5 in the midgut express Ret whereas only a minor population of Pax2Cre-derived cells does so. This further supports the idea that Pax2Cre and Wnt1Cre delineate molecularly distinct populations of early migratory ENS progenitor cells.

We next evaluated expression of markers in normal P9 pups that were also lineage labeled with Pax2Cre or Wnt1Cre crossed with *Rosa26* reporters. First, based on morphology and expression of the pan-neuronal marker Hu, approximately half of colonic myenteric ganglion cells were labeled with either lineage marker (*Figure 3—figure supplement 1a, b, g*, pie chart). Because the sum of the

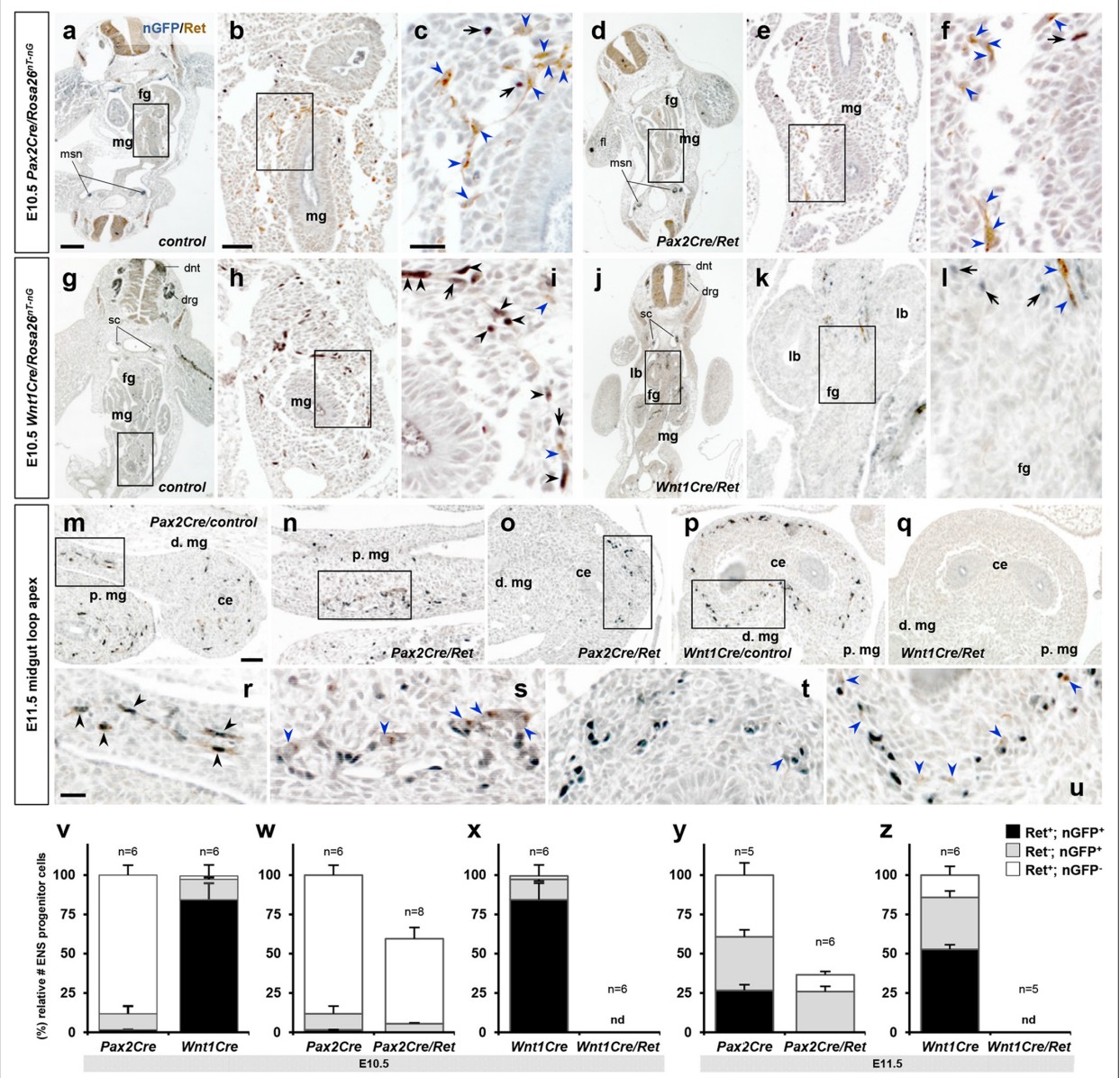

**Figure 2.** Developmental transition of Ret expression in migratory enteric nervous system (ENS) progenitor cells. (**a–u**) Histological sections prepared from E10.5 *Pax2Cre/Ret/Rosa26^{nT-nG}* mutant (**d–f**), *Wnt1Cre/Ret/Rosa26^{nT-nG}* mutant (**j–l**), and their littermate controls (a–c and g–l, respectively), and E11.5 *Pax2Cre/Ret/Rosa26^{nT-nG}* mutant (**n–o**), *Wnt1Cre/Ret/Rosa26^{nT-nG}* mutant (**q**), and their littermate control embryos (m and p, respectively) stained for nuclear-GFP (blue) and Ret (brown). Magnified views of the bracketed areas (midgut) in a, d, g, and j are shown in b, e, h, and k, respectively; of the bracketed areas in b, e, h, and k are shown in c, f, I, and l, respectively; of the bracketed areas in m–p are shown in r–u, respectively. Black arrows denotes lineage marker nGFP⁺ nuclei with no Ret stain, black arrowheads denoted cells that are positive for both nuclear-GFP and Ret, and blue arrowheads denote non-Cre (nGFP⁻) Ret⁺ cells along the gut walls. (**v–z**) Compiled representations of relative fractions of Ret⁺, nGFP⁺ (black); Ret⁻, nGFP⁺ (gray); and Ret⁺, nGFP⁻ (white) ENS progenitor cells comparing (**v**) E10.5 midgut areas of *Pax2Cre/Rosa26^{nT-nG}* (n=6; 249.7±15.7 cells/embryo) and *Wnt1Cre/Rosa26^{nT-nG}* (n=6; 178.8±11.4 cells/embryo) embryos, (**w**) E10.5 midgut areas of *Pax2Cre/Ret/Rosa26^{nT-nG}* mutants (n=8, 146.9±3.6 cells/embryo) and littermate control embryos, (**x**) E10.5 midgut areas of *Wnt1Cre/Ret/Rosa26^{nT-nG}* mutants (n=6, no nGFP⁺ or Ret⁺ cell was found) and littermate control embryos, (**y**) E11.5 midgut loop apex areas of *Pax2Cre/Ret/Rosa26^{nT-nG}* mutants (n=6, 301.9±35.9 cells/embryo) and littermate control embryos (n=5, 826.2±32.2 cells/embryo), and (**z**) E11.5 midgut loop apex areas of *Wnt1Cre/Ret/Rosa26^{nT-nG}* mutants (n=4, no nGFP⁺ or Ret⁺ cell was found) and littermate control embryos (n=6, 943.8±63.8 cells/embryo). Data in v are the control mouse groups in w and x, shown together for easier comparison. Ret⁻, nGFP⁻ (double negative) cell populations were not included in this analysis. Error bars; mean ± s.e.m. nd = not detected. Abbreviations: dnt, dorsal neural tube; drg, dorsal root ganglia; msn, mesonephros; sc, sympathetic chain ganglia. Scale bars, 200 µm (**a, d, g, j**), 50 µm (**b, e, h, k, m–q**), 20 µm (**c, f, i, l, r–u**).

The online version of this article includes the following source data for figure 2:

*Figure 2 continued on next page*

*Figure 2 continued*

**Source data 1.** Raw data for panels v to x and panels y and z (numbers and fractions of linegae-labeled Ret+, lineage-labled Ret-, and lineage-unlabeled Ret+ ENS progenitor cells).

two individual estimates is close to 100%, this is consistent with the proposition that all or most ENS neurons are derived from either a Pax2Cre or a Wnt1Cre lineage and that the two Cre lines are mostly if not fully nonoverlapping in their recombination domains. That is, if there was an appreciable degree of overlap between the two lineages, the same fraction of Hu$^+$ neurons would have to be derived from a third source, for which there is no evidence.

As summarized earlier, colonic peristalsis is facilitated by an integrated circuitry from mechanosensory neuron to interneuron to motor neuron to muscle contraction (*Figure 3a*). Electrophysiology studies of guinea pig colon showed that mechanosensory neurons release the excitatory neurotransmitter calcitonin gene-related peptide (CGRP) to initiate peristaltic contractions (*Sarna, 2010*). By immunostaining, we identified that CGRP$^+$ neurons in the postnatal P9 mouse distal colon are located in the myenteric plexus and send their projections to the gut luminal epithelium, consistent with a mechanosensory fate (*Figure 3b*). Nitric oxide synthase (NOS) is recognized as a marker for both inhibitory interneurons and inhibitory motor neurons (*Costa et al., 1992*; *Timmermans et al., 1994*; *Sang and Young, 1996*). In the mouse P9 colon, NOS$^+$ neurons are found in the myenteric ganglia and their axons are found exclusively within the circular muscle layer (*Figure 3c*), implying that these represent at least a subset of motor neurons. Interneurons are not readily identifiable with this circular cross-sectional analysis because of their exclusively interganglionic (mostly ascending/descending) projections.

To discern the lineage origins of these mature cell types, we evaluated the expression of these markers in lineage-labeled P9 pups. Only approximately one-third of neurons of both lineages in the distal colon were CGRP$^+$ or NOS$^+$ (*Figure 3—figure supplement 1g*, bar graph), which means that additional neuronal fates (white bars in *Figure 3—figure supplement 1g*) are taken by both lineages (which is expected). Less than 10% of myenteric ganglion neurons were CGRP$^+$ (*Figure 3—figure supplement 1h*, pie chart). Strikingly, the vast majority (>90%) of these were Pax2Cre-derived (*Figure 3d*, *Figure 3—figure supplement 1c, d, h*, bar graph). A minimum number (approx. 3%) of CGRP$^+$ neurons were labeled by Wnt1Cre (*Figure 3f*, *Figure 3—figure supplement 1, 1h*, bar graph), which indicates that Wnt1Cre-derived cells have little or no commitment to a mechanosensory fate in the distal colon. Approx. 25% of myenteric ganglion neurons were NOS$^+$. The majority of these were derived from the Wnt1Cre lineage (*Figure 3g*, *Figure 3—figure supplement 1f, h*, bar graph), although Pax2Cre-derived neurons also gave rise to a NOS-reactive population of the distal colon (*Figure 3e*, *Figure 3—figure supplement 1e, h*, bar graph). As mentioned above, NOS identifies both inhibitory interneurons and inhibitory motor neurons in the intestine, so these results do not by themselves indicate if Pax2Cre-labeled NOS$^+$ and Wnt1Cre-labeled NOS$^+$ neurons are the same or different fates (this point is addressed below in lineage-specific *Ednrb* mutants).

If Pax2Cre and Wnt1Cre label distinct subsets of ENS neurons in normal mice, a prediction is that these subsets might be selectively ablated in lineage-specific *Ednrb* mutant mice. To address this, we immunohistochemically analyzed axonal projections from the myenteric plexus in serial sections of the colons of both lines of mutant mice at P9. Importantly, sections were taken at the transition between ganglionic and aganglionic areas (*Figure 3—figure supplement 3c, d, g, h*). Adjacent sections were stained for CGRP or NOS. In this transition zone, we observed a profound reduction of CGRP$^+$ axons projecting toward the colonic lumen in *Pax2Cre/Ednrb* mutant guts (*Figure 3j and p*), equal to that in global *Ednrb* mutants (*Figure 3i and p*), whereas NOS$^+$ axons in the circular muscle were present normally in the adjacent section (*Figure 3n and q*). There were some CGRP$^+$ axons projecting longitudinally along the submucosal layer (*Figure 3j*, asterisks), although these most likely originate from sacral neural crest-derived pelvic ganglion neurons (*Figure 3—figure supplement 2*) or trunk neural crest-derived DRG neurons (*Wolfson et al., 2023*), and were also detected in the global *Ednrb* mutant gut (*Figure 3i*, asterisks). *Wnt1Cre/Ednrb* mutant guts exhibited abundant CGRP$^+$ axons projecting toward sensory epithelium (*Figure 3k and p*) while their circular muscle layers showed a total loss of NOS$^+$ axonal innervation (*Figure 3o and q*). NOS$^+$ neurons were still present in the myenteric plexus of the *Wnt1Cre/Ednrb* mutant distal colon (*Figure 3o*, inset arrowhead); as these neurons project neither to sensory nor motor targets, we suspect that these are interneurons and likely originate

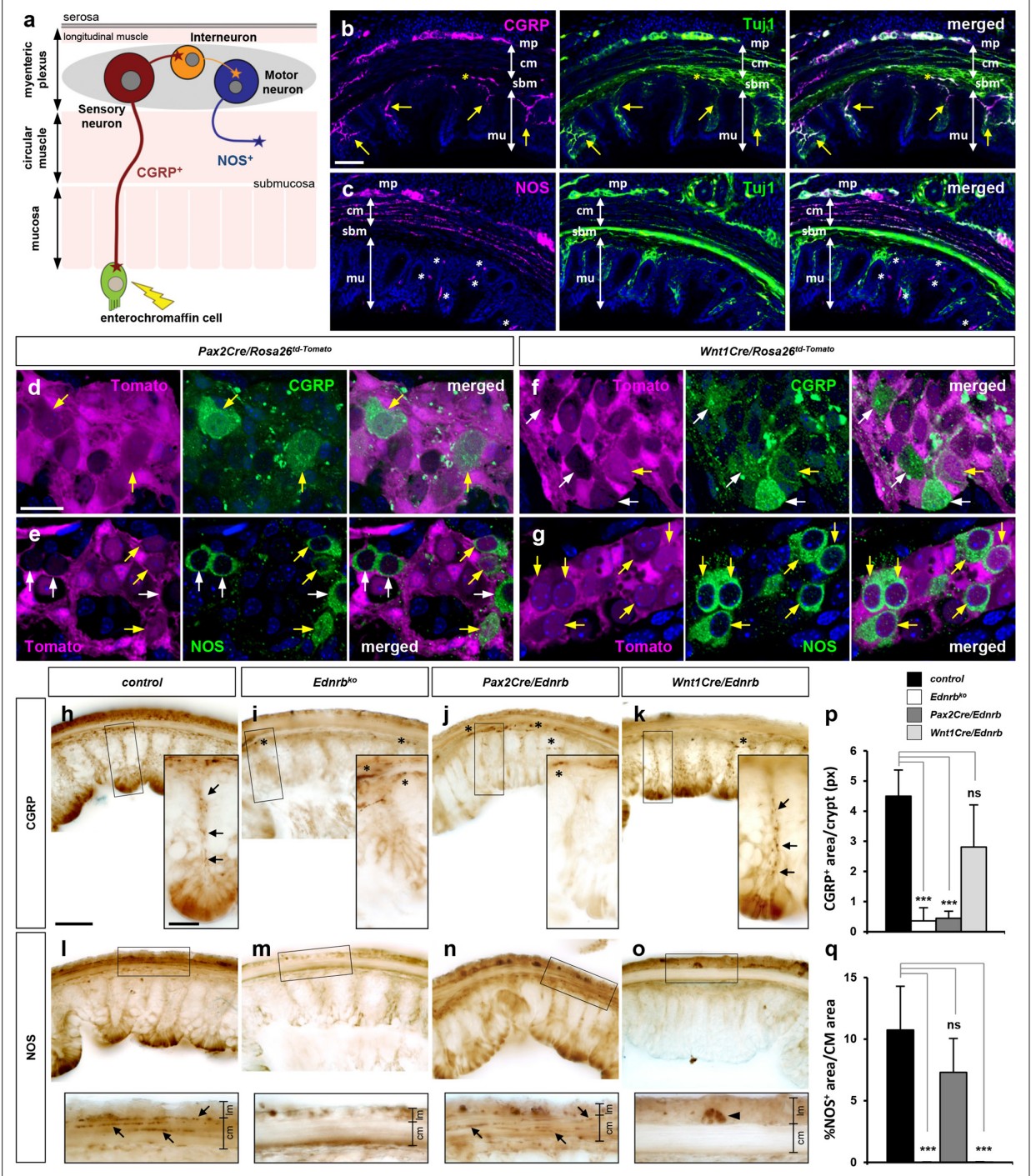

**Figure 3.** Mechanosensory fate of *Pax2Cre* lineage and selective loss of CGRP[+] myenteric mechanosensory axons in *Pax2Cre/Ednrb* mutant distal colon. (**a**) Schematic illustration of neuronal subtypes in the colonic myenteric ganglia and their end organs. Mechanosensory neurons (red) extend their peripheral axons to the mechanosensory (enterochromaffin) cells while their other axons synapse to interneurons (orange). Motor neurons (blue) innervate the colonic circular muscle to generate colonic peristalsis. (**b, c**) Confocal images of transverse sections of the distal colon co-immunostained for CGRP and Tuj1 (**b**) or NOS and Tuj1 (**c**) show selective label of sensory and motor axons by CGRP (**b**) and NOS (**c**), respectively. Yellow arrows in b denote CGRP[+] axons that project through each crypt to reach enteric epithelium. Yellow asterisks in b indicate CGRP[+] axons from pelvic ganglia along the submucosal layer. White asterisks in c point to NOS-reactive non-neuronal epithelial cells of the mucosa. (**d–g**) Confocal images of the myenteric ganglia of P9 distal colons isolated from *Pax2Cre/Rosa26[td-Tomato]* (**d, e**) and *Wnt1Cre/Rosa26[td-Tomato]* (**f, g**) mice immunostained for CGRP (**d, f**) and NOS (**e, g**). Low-magnification views from which these images were taken are shown in **Figure 4**. Compared to b and c, this wholemount preparation emphasizes cell bodies in the myenteric plexus rather than axon projections. Yellow arrows point to lineage marker-positive CGRP[+] or NOS[+] neurons, whereas

*Figure 3 continued on next page*

*Figure 3 continued*

white arrows indicate lineage marker-negative CGRP⁺ or NOS⁺ neurons of the myenteric ganglia. (**h–o**) Serial transverse vibratome sections of the transition areas (see *Figure 6c and h*) of the distal colons isolated from P9 control (**h, l**), global *Ednrb* (**i, m**), *Pax2Cre/Ednrb* (**j, n**), and *Wnt1Cre/Ednrb* (**k, o**) mutants immunostained for CGRP (**h–k**) or NOS (**l–o**). Magnified views of bracketed areas are shown in the insets. Arrows in h and k denote CGRP⁺ axonal projections toward mucosal epithelium in a single crypt. Arrows in l and n denote NOS⁺ axonal projections perforating throughout the circular muscle layer. Arrowhead in o indicates a NOS⁺ presumed myenteric interneuron (because it lacks motor axons in the circular muscle layer) of *Wnt1Cre/Ednrb* mutant colon. Scale bars, 50 µm (**b, c, h–o**), 20 µm (**d–g, h–o** insets). (**p**) Complied representation of the average pixel values of CGRP⁺ axon area per single crypt. (**q**) Complied representation of the average fraction of NOS⁺ axon area in the circular muscle area. Analysis includes 24 crypts from 8 distal colon sections per genotype (four controls, two *Ednrb*, three *Pax2Cre/Ednrb,* and three *Wnt1Cre/Ednrb* mutants). Error bars; mean±s.d. p-Values: ***p<0.001, ns = not significant. Abbreviations: cm, circular muscle; lm, longitudinal muscle; mp, myenteric plexus; mu, mucosa; sbm, submucosa.

The online version of this article includes the following source data and figure supplement(s) for figure 3:

**Source data 1.** Raw data for panel p (CGRP+ pixel area/crypt) and panel q (NOS+ pixel area/circular muscle ROI area).

**Figure supplement 1.** Pax2Cre-derived cells preferentially give rise to CGRP⁺ myenteric neurons of the distal colon.

**Figure supplement 1—source data 1.** Raw data for panels g and h (numbers and fractions of of lineage labeled, CGRP+ and/or NOS+ neurons in the ENS of the distal colon).

**Figure supplement 2.** Pelvic ganglion neurons and their projections express CGRP.

**Figure supplement 3.** Transition areas in lineage-specific *Ednrb* mutant distal colon also contain lineage-unlabeled neurons.

from the Pax2Cre lineage. Thus, in the transition area, defective migration of Pax2Cre-derived enteric progenitor cells results in selective loss of myenteric mechanosensory neurons, while myenteric motor neurons and their innervations of the circular muscle were not affected. Reciprocally, defective migration of Wnt1Cre-derived ENS progenitors results in selective loss of motor neurons with no impact on the mechanosensory neuron population.

These results collectively demonstrate that Pax2Cre-labeled and Wnt1Cre-labeled cells have a distinct and nonoverlapping marker expression program during their embryonic migration, and have a distinct set of terminal differentiation fates in the postnatal distal colon. A further conclusion from the analysis of conditional *Ednrb* mutants is that neither lineage can modulate its normal fate to compensate for the absence of the other.

## Pax2Cre lineage-specific *Ret* mutant embryos exhibit defective enteric progenitor migration

ENS development requires a functional GDNF-Ret signaling axis alongside that of Edn3-Ednrb. The unexpected observation that Ednrb function is required in both the Wnt1Cre and Pax2Cre domains in ENS development led us to also address Ret function in these lineages. A previous study (*Luo et al., 2007*) documented Hirschsprung disease phenotypes in mice with conditional Ret mutation in the Wnt1Cre domain (*Wnt1Cre/Ret* mutants), including lethality around postnatal day 21. Global *Ret* deficiency results in neonatal lethality because of renal agenesis (*Schuchardt et al., 1994*; *Durbec et al., 1996*), and because Pax2Cre is expressed in intermediate mesoderm, *Pax2Cre/Ret* mutants in our colony also died in the early postnatal period. We therefore explored the consequences of *Ret* deficiency in the Pax2Cre lineage during embryonic enteric progenitor cell migration. In *Pax2Cre/Ret/Rosa26^{lacZ}* mutant embryos at E12.5, we observed an obvious deficiency of Pax2Cre-labeled cells migrating throughout the midgut, and no Pax2Cre-labeled cells migrated caudally beyond the cecum (*Figure 4b*), whereas lineage-labeled cells reached the terminal hindgut in littermate controls (*Figure 4a*). This result implies a function for Ret within the Pax2Cre lineage. This was surprising because the expression analysis in normal embryos described above indicated that so few (1.6%) of the ENS progenitors at E10.5 are of the Pax2Cre lineage and are Ret⁺ (*Figure 2a–c and v*).

We considered whether Ret expression in the Pax2Cre lineage changes around E10.5 that might account for these observations. We found that a substantially larger fraction of Pax2Cre-labeled cells in the distal midgut (at the migratory wavefront) in control E11.5 *Pax2Cre/Rosa26^{nT-nG}* embryos expressed Ret (*Figure 2m and y*, black bar) relative to E10.5 (*Figure 2w*, black bar). This demonstrates that the Pax2Cre-labeled population in control embryos expands significantly during the E10.5–11.5 period. This could occur by proliferation of the very small Ret⁺ population that was present at E10.5, or could occur by conversion of Ret⁻ to Ret⁺ cells within the lineage, or both, and perhaps was already

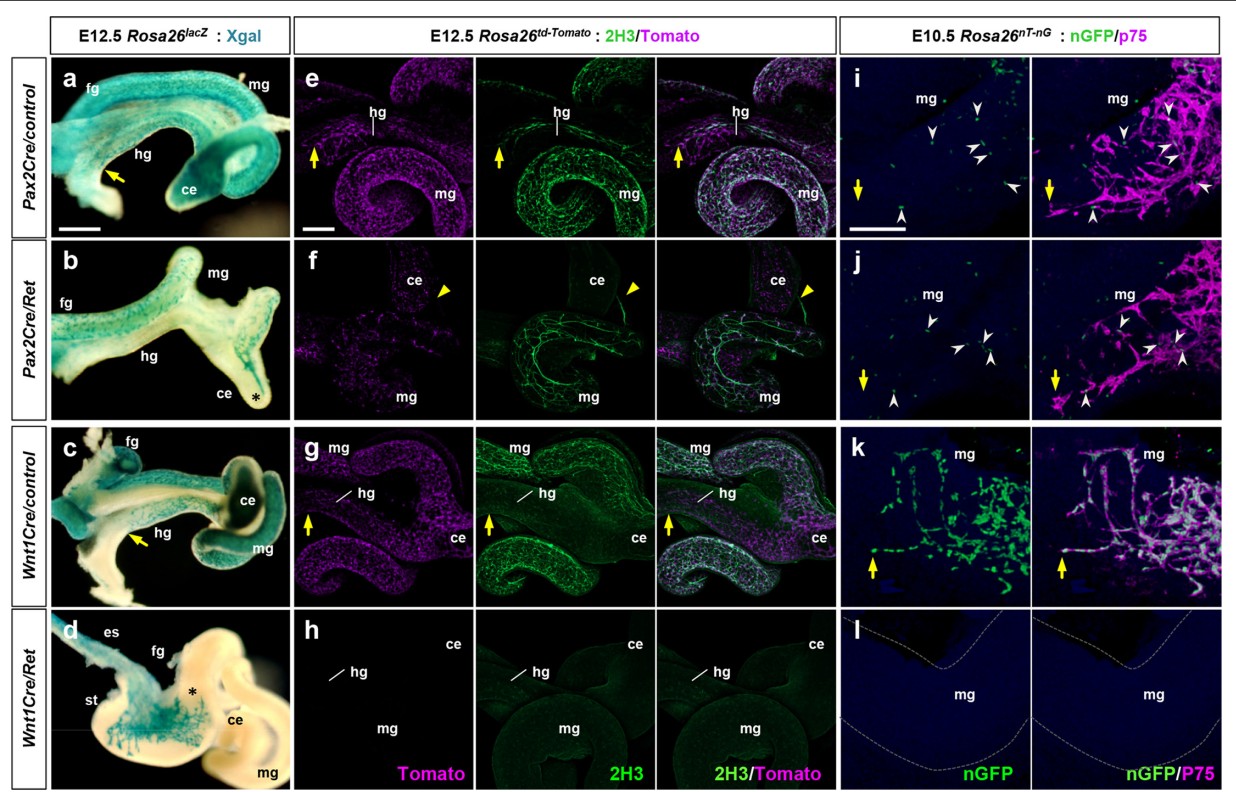

**Figure 4.** Pax2Cre-specific deletion of *Ret* results in defective enteric nervous system (ENS) progenitor migration and differentiation. (**a–d**) Wholemount preparations of E12.5 guts isolated from *Pax2Cre/Ret/Rosa26^lacZ* (**b**), *Wnt1Cre/Ret/Rosa26^lacZ* (**d**) mutant embryos and their littermate controls (a and c, respectively) stained with Xgal (blue). Arrows denote the wavefronts of lineage-labeled cells in the hindguts of the controls. Asterisks denote halted migration of *Ret*-deficient *Pax2Cre*-labeled cells at the cecum (**b**) and of *Ret*-deficient *Wnt1Cre*-labeled cells at the caudal end of the stomach (**d**). (**e–h**) Wholemount preparations of E12.5 guts isolated from *Pax2Cre/Ret/Rosa26^td-Tomato* (**f**), *Wnt1Cre/Ret/Rosa26^td-Tomato* (**h**) mutant embryos and their littermate controls (e and g, respectively) stained for neuronal 2H3 (green). Arrows denote the wavefronts of lineage-labeled cells in the hindguts of the control embryos. Arrowheads denote halted migration of *Ret*-deficient *Pax2Cre*-labeled cells at the cecum (**f**). (**i–l**) Wholemount preparations of E10.5 guts isolated from *Pax2Cre/Ret/Rosa26^nT-nG* (**j**), *Wnt1Cre/Ret/Rosa26^nT-nG* (**l**) mutant embryos and their littermate controls (i and k, respectively) stained for nuclear-GFP (green) and ENS progenitor marker p75 (magenta). Arrows denote the wavefronts of lineage-labeled cells in the midguts. Arrowheads denote p75$^+$ *Pax2Cre*-labeled cells at the midgut areas (**i, j**). Abbreviations: es, esophagus; st, stomach. Scale bars, 500 μm (**a–d**), 250 μm (**e–h**), 50 μm (**i–l**).

just underway at E10.5. Obviously, at both E10.5 and E11.5, there were no Ret$^+$ cells in the Pax2Cre lineage in *Pax2Cre/Ret* mutants (***Figure 2n, o, w, and y***).

We next performed wholemount staining with the neural progenitor marker p75 (at E10.5) or with the pan-neuronal marker 2H3 (at E12.5) on guts isolated from *Pax2Cre/Ret/Rosa26^td-Tomato* mutants and their littermate controls. This allowed us to visualize the Pax2Cre lineage in *Pax2Cre/Ret* mutant embryos alongside other neuronal lineages (i.e. the Wnt1Cre lineage) in which Ret function is intact. At E10.5, p75-expressing enteric progenitors included Pax2Cre lineage cells co-migrating with non-Pax2Cre cells (***Figure 4i***), the latter which presumably consists of Wnt1Cre-derived cells. In *Pax2Cre/Ret* mutant embryos at this stage, p75-expressing cells again included those with and without the Pax2Cre lineage label, demonstrating a normal caudal progression (***Figure 4j***). Histological Ret expression analysis in *Pax2Cre/Ret/Rosa26^nT-nG* embryos clearly showed that some migratory progenitor cells express Ret (***Figure 2d–f***). Since *Ret* is knocked out in *Pax2Cre/Ret* mutants, these Ret$^+$ cells are presumed to be Wnt1Cre-derived. Quantitative evaluation, however, indicated that overall ENS progenitor cell number was already affected as early as E10.5 in the *Pax2Cre/Ret* mutant background (***Figure 2w***): both the Pax2Cre-derived cell population and the Ret-expressing non-Pax2Cre population were reduced by half, and therefore the total number of countable cells (expressing one or both markers) per midgut from each mutant embryo was also reduced by half. A somewhat larger reduction in total countable cells was observed in *Pax2Cre/Ret* mutants at E11.5 (***Figure 2y***) compared

to E10.5 (*Figure 2w*). Consequently, at E12.5, compared to control littermate (*Figure 4e*), *Pax2Cre/Ret* mutants showed a marked deficiency of 2H3-positive cells through the midgut loop, and no 2H3 signal was detected caudal to the cecum (*Figure 4f*). The interpretation of this analysis is that *Ret* deficiency in the Pax2Cre lineage impacts ENS progenitor cells derived from the Pax2Cre lineage as well as from the presumptive Wnt1Cre lineage, starting as the migratory cells enter the midgut.

In parallel we also crossed the conditional *Ret* allele into the *Wnt1Cre/Rosa26* background. Wnt1Cre-labeled cells at E12.5 populated the esophagus and stomach but failed to migrate into the intestine (*Figure 4d*), which recapitulates the global *Ret* mutant phenotype (*Schuchardt et al., 1994*; *Durbec et al., 1996*). Wholemount 2H3 staining confirmed the total loss of enteric progenitor cells along the entire axis of the developing intestine (*Figure 4h*). Defective progenitor entry into the intestinal tract was evident as early as E10.5 based on wholemount p75 staining (*Figure 4i*) as well as Ret expression analysis on histological sections prepared from *Wnt1Cre/Ret/Rosa26*$^{nT-nG}$ embryos at E10.5 (*Figure 2j–l and x*) and E11.5 (*Figure 2q and z*). It should be noted that our analysis at E10.5–12.5 precedes the E14.5 arrival of neurogenic Schwann cell precursors, which are neural crest-derived but which do not require Ret for their migration (*Uesaka et al., 2015*). These results demonstrate that Ret function in the Wnt1Cre domain is absolutely required for migration of all enteric progenitors, Wnt1Cre-derived as well as non-Wnt1Cre-derived (i.e. Pax2Cre-derived), into and through the intestinal tract.

To better understand how GDNF-Ret signaling influences the Pax2Cre and Wnt1Cre lineage cells, we employed collagen gel explant cultures of serial 200 μm slices prepared from E10.5 midgut segments collected in a rostrocaudal sequence (*Figure 5—figure supplement 1a*). After 3 days in culture we visualized td-Tomato as a lineage marker, stained for Hu as a neuronal marker, and BLBP as a glial marker. No outgrowth was observed in the absence of added factors (*Figure 5a*). Hu and BLBP were both detected within the untreated explanted gut slices (*Figure 5a*), showing that enteric progenitor cells present in the gut slices survived for 3 days in vitro and sustained a normal differentiation program without growth factors. GDNF induced robust radial outgrowth from control midgut slices, which included a mixed population of neurons and glia; Pax2Cre lineage cells were present in these outgrowths (*Figure 5b*). Control explants showed a progressively diminishing response along the rostrocaudal axis (*Figure 5g*), which reflects the number of enteric progenitor cells present in each slice. *Pax2Cre/Ret* mutant gut explants, however, exhibited little or no outgrowth in response to GDNF (*Figure 5c and g*), even though lineage-positive, Hu-positive, and BLBP-positive cells are present in the explant itself. Importantly, Ret-expressing non-Pax2Cre lineage (i.e. likely Wnt1Cre-derived) cells are present throughout the midgut at E10.5 (*Figure 2f and w*), yet there were no lineage negative outgrowths in response to GDNF in *Pax2Cre/Ret* explants (*Figure 5c*). This indicates that the Wnt1Cre-derived cells in the midgut do not harbor a cell-autonomous migratory response to GDNF, but instead require co-migration with Pax2Cre lineage cells.

Control E10.5 *Wnt1Cre/Rosa26*$^{td-Tomato}$ midgut explants demonstrated response to GDNF at each axial level (*Figure 5d, e and h*), similar to *Pax2Cre/Rosa26*$^{td-Tomato}$ explants. *Wnt1Cre/Ret* mutant gut explants showed no lineage marker, neuronal Hu, or glial BLBP-positive cells along the entirety of the midgut segments, within or outgrown from the explant (*Figure 5f and h*), confirming the total absence of all enteric progenitor cells, including Pax2 lineage cells, in this mutant background (*Figure 4d*). Collectively, we interpret these results to show that ENS progenitor migration is an orchestrated process that requires GDNF response in both Pax2Cre and Wnt1Cre progenitor populations. To support migration, each progenitor lineage requires migratory function (i.e. Ret function) in the other, albeit at different times and locations along the gut axis: Ret deficiency in the Wnt1Cre lineage blocks migration of both lineages at the duodenum, whereas Ret deficiency in the Pax2Cre lineage blocks migration of both lineages at the hindgut.

## Divergent lineage-specific regulation of Ret signaling by Edn3-Ednrb

Edn3-Ednrb and GDNF-Ret signaling are both required for caudal migration of Wnt1Cre-derived and Pax2Cre-derived enteric progenitors in vivo. We used the gut slice explant assay to address the relationship between these signaling mechanisms. Midgut slices from *Pax2Cre/Rosa26*$^{td-Tomato}$ and *Wnt1Cre/Rosa26*$^{td-Tomato}$ embryos were cultured in collagen beds supplemented with EDN3 or with the p75 ligand NGF for 3 days in vitro. While explanted gut sections at E10.5 contain cells that express p75 and Hu, neither EDN3 nor NGF promoted enteric progenitor cell outgrowth compared to the

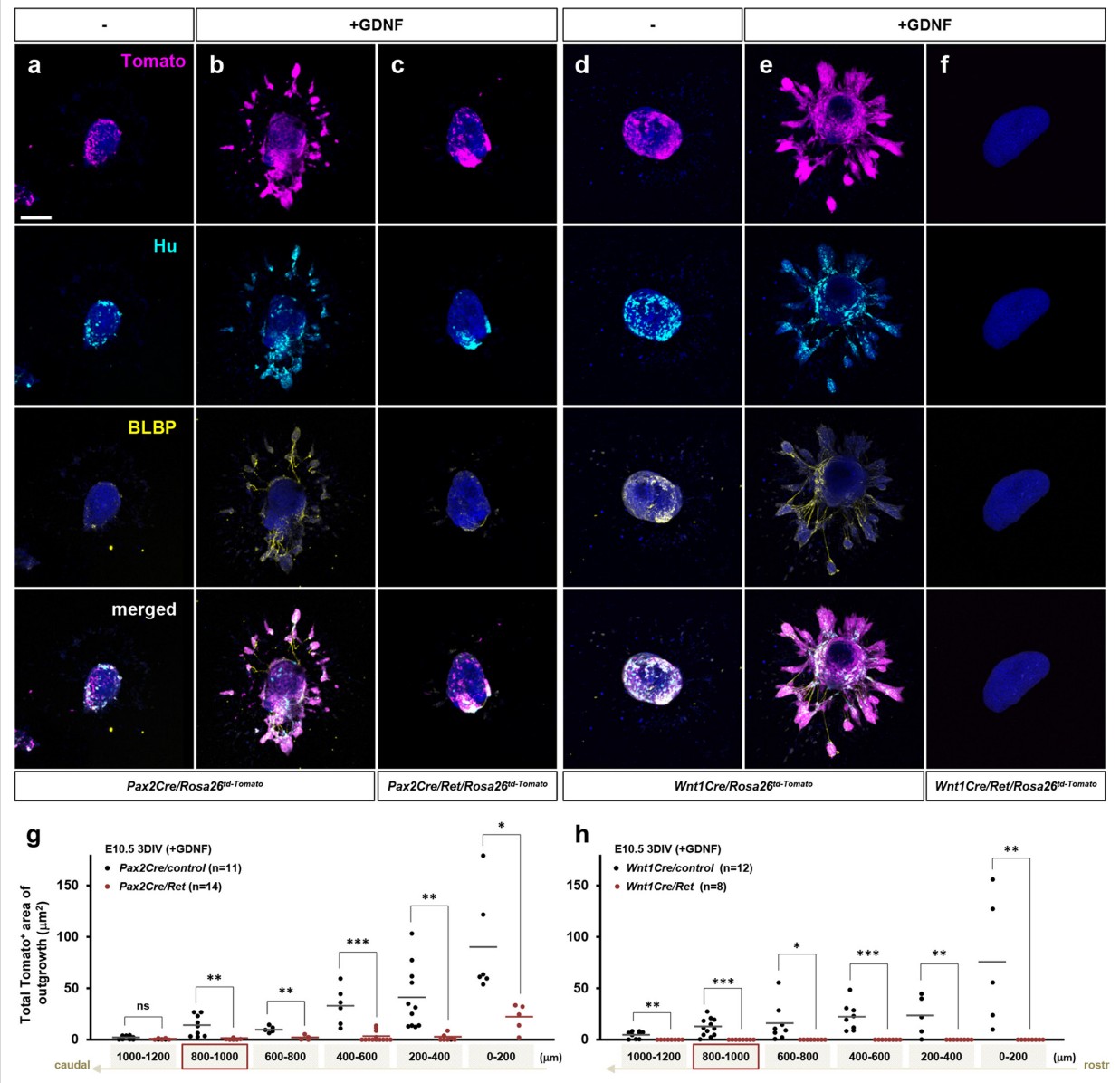

**Figure 5.** Defective GDNF-induced outgrowth and differentiation of *Pax2Cre/Ret* mutant enteric nervous system (ENS) progenitors in vitro. (**a–f**) Confocal images of 200 mm midgut sections isolated from an E10.5 *Pax2Cre/Ret/Rosa26^td-Tomato* mutant (**c**) and its littermate control (**a, b**), and a *Wnt1Cre/Ret/Rosa26^td-Tomato* mutant (**f**) and its littermate control (**d, e**) cultured 3 days in vitro (3DIV) in the absence (**a, d**) or presence of GDNF (**b, c, e, f**), then immunolabeled for neuronal Hu (cyan) and glial BLBP (yellow). Scale bars, 250 μm. (**g**) Complied representation of the total area of outgrowth (Tomato⁺ area around the explanted gut tube) from *Pax2Cre/control* (black dots; n=11) and *Pax2Cre/Ret* mutant (red dots; n=14) embryos in the presence of GDNF. (**h**) Compiled representation of the total outgrowth area from *Wnt1Cre/control* (black dots; n=12) and *Wnt1Cre/Ret* mutant (red dots; n=8) embryos in the presence of GDNF. See *Figure 5—figure supplement 1a* for rostrocaudal orientation of the midgut slice preparations. Each dot represents the outgrowth of each gut slice prepared from each embryos used for data acquisition (lost individual midgut sections are not included as data points). The images shown in a–f represent sections that are 800–1000 μm caudal to the foregut-midgut junction (as indicated by the red boxes in g–h). p-Values: *p<0.05, **p<0.01, ***p<0.001, ns = not significant.

The online version of this article includes the following figure supplement(s) for figure 5:

**Figure supplement 1.** Schematic illustrations of rostrocaudal orientation of gut slice explant culture.

**Figure supplement 2.** EDN3 exhibited marginal effects on E10.5 enteric progenitor outgrowth in vitro.

activity of GDNF (*Figure 5—figure supplement 2*). Similarly, at E11.5, EDN3 and NGF induced only minimal outgrowth from both Pax2Cre- and Wnt1Cre lineage-labeled midgut sections (*Figure 6—figure supplement 1*). Impaired cell migration in vivo (*Figure 1f and h*) has at various times been previously interpreted in global *Edn3* and *Ednrb* mutants to represent a chemoattractive role for Edn3-Ednrb signaling, but as previously shown in vitro (*Barlow et al., 2003*; *Goto et al., 2013*; *Nagy and Goldstein, 2017*) and in our results, Edn3-Ednrb signaling does not by itself induce migration of either Pax2Cre-labeled or Wnt1Cre-labeled enteric progenitors.

The in vitro explant assays clearly demonstrate that GDNF-Ret signaling is the major mechanism that supports ENS progenitor migration. To address why enteric progenitor migration in vivo is defective in *Ednrb*-deficient backgrounds, we performed gut slice explant assays using E11.5 midguts isolated from *Pax2Cre/Ednrb/Rosa26^{td-Tomato}* and *Wnt1Cre/Ednrb/Rosa26^{td-Tomato}* mutant embryos. Across the rostral-caudal axis of the midgut, *Pax2Cre/Ednrb* mutant explants showed a blunted response to GDNF compared to control cells (*Figure 6a and c*), which was more severe at the distal end (*Figure 6b, d, i, and j*), even though lineage-positive cells were present. *Wnt1Cre/Ednrb* mutant midgut segments exhibited a similar trend as *Pax2Cre/Ednrb* mutant midgut in both proximal (*Figure 6e, g, and k*) and distal segments (*Figure 6f, h, and l*), including greater impairment of GDNF responsiveness at the migratory wavefront. These results show that the Pax2Cre and Wnt1Cre lineages both rely on endothelin signaling to fully populate the gut with GDNF-responsive cells, and imply that endothelin signaling is needed to support GDNF responsiveness in enteric progenitors.

Next we asked whether *Ednrb* deficiency alters Ret expression in the enteric progenitor cell population. We used two-color IHC for nuclear-GFP and Ret on histological sections prepared from E11.5 *Pax2Cre/Ednrb/Rosa26^{nT-nG}* mutant embryos and their littermate controls. As also noted earlier (*Figure 2m and y*), in normal E11.5 embryos a large fraction of Pax2Cre-derived cells express Ret at the migratory wavefront (*Figure 7a and e*). In the *Pax2Cre/Ednrb* mutant, however, Pax2Cre-labeled cells populating the apex of the midgut loop were Ret-negative, and the surrounding Ret-expressing cells were mostly unlabeled by the Pax2Cre lineage marker (*Figure 7b and f*). We measured the number of nGFP-positive only, Ret-positive only, and nGFP and Ret double positive cells at three different axial levels: distal midgut, caudal half of the proximal midgut (toward the cecum; equivalent to 0–600 μm in *Figure 6i*), and rostral half of the proximal midgut (toward the foregut-midgut junction; equivalent to 600–1200 μm in *Figure 6i*). The analysis confirmed that absence of *Ednrb* resulted in far fewer Pax2Cre lineage-derived cells expressing Ret in the more caudal segments (*Figure 7i*). Thus, the expansion of the Ret$^+$ subpopulation of the Pax2Cre lineage that occurs between E10.5 and 11.5 (*Figure 2m and y*) requires Ednrb signaling in a cell-autonomous manner. As noted earlier, this could occur via proliferation of the very small Ret$^+$ population at E10.5, by induction of Ret expression in previously Ret-negative cells, or both. The important conclusion is that this process by Pax2Cre-derived cells is dependent on Ednrb.

We performed the same analysis for E11.5 *Wnt1Cre/Ednrb/Rosa26^{nT-nG}* mutants and their littermate embryos (*Figure 7c, d, g, h, and j*). Unlike the observation in the Pax2Cre lineage, Edn3-Ednrb signaling does not affect Ret expression in Wnt1Cre-derived migratory enteric progenitor cells. Nonetheless, these cells are unable to respond to GDNF in vivo (*Figure 1h*) and in vitro (*Figure 6g and h*).

In general, extracellular ligands of receptor tyrosine kinases (RTKs) trigger autophosphorylation of intracellular tyrosine residues that ultimately recruit intracellular adaptor proteins to initiate downstream signaling. In a variety of ways, RTK signaling can require mediation by GPCRs, a mechanisms called transactivation (*Kilpatrick and Hill, 2021*). Prior in vitro explant studies have implicated a convergence of Ret and Ednrb at the level of protein kinase A activity (*Barlow et al., 2003*; *Goto et al., 2013*), although other mechanisms of transactivation have been described for Ednrb to other RTKs (*Grantcharova et al., 2006*; *Harun-Or-Rashid et al., 2016*), and Ret with other GPCRs (*Tsuchioka et al., 2008*). Because Ret expression was preserved in Wnt1Cre-derived enteric progenitors even as they were nonresponsive to GDNF, we considered that transactivation via Ednrb might be involved. To address this, we fluorescently visualized the phosphorylation status of Ret (i.e. its degree of activation) in Wnt1Cre-derived cells in the midgut loop of E11.5 *Wnt1Cre/Ednrb/Rosa26^{nT-nG}* mutants and their littermate embryos. Confocal z-stack images showed that Ret protein was detected in both the cytoplasm and nucleus of Wnt1Cre-labeled migratory cells (*Figure 7k–n*, cyan; nuclear localization of Ret and many other RTKs has been previously noted *Chen and Hung, 2015*; *Chen et al., 2020*; and is not unusual). An antibody specific for phospho-Ret Y1015 was detected in the majority (87%) of

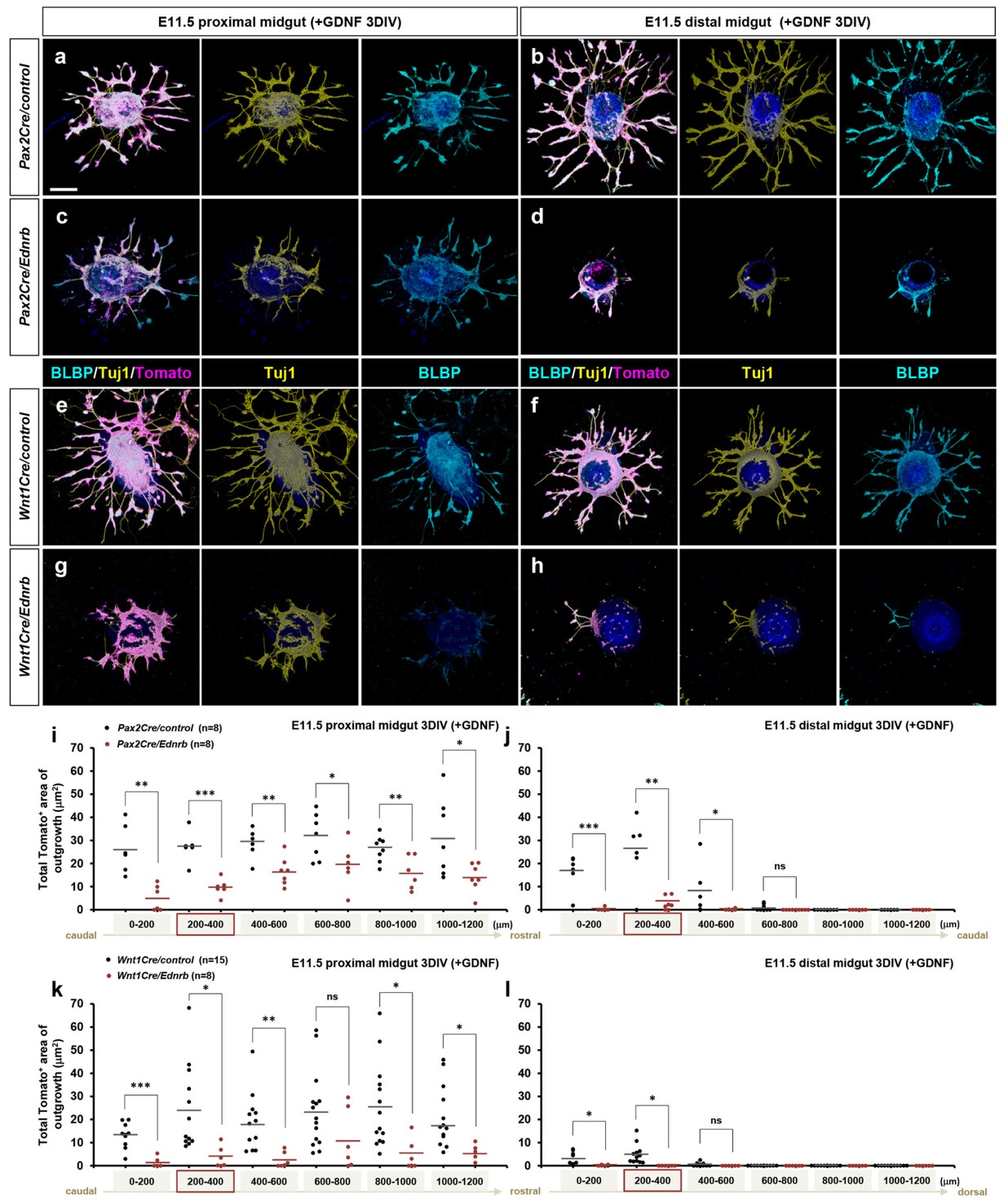

**Figure 6.** *Ednrb*-deficient enteric nervous system (ENS) progenitor cells populating the migratory wavefront exhibit attenuated response to GDNF in vitro. (**a–h**) Confocal z-stack images of 200 µm proximal (**a, c, e, g**) and distal midgut (**b, d, f, h**) sections isolated from an E11.5 *Pax2Cre/Ret/Rosa26^td-Tomato* mutant (**c, d**) and its littermate control (**a, b**), and a *Wnt1Cre/Ret/Rosa26^td-Tomato* mutant (**g, h**) and its littermate control (**e, f**) cultured 3 days in vitro (3DIV) in the presence of GDNF and then immunolabeled for neuronal Tuj1 (yellow) and glial BLBP (cyan). Scale bars, 250 µm. (**i, j**) Compiled representation of the total area of outgrowth (Tomato⁺ area around the explanted gut tube) from proximal and distal midguts of *Pax2Cre/control* (black dots; n=8) and *Pax2Cre/Ret* mutant (red dots; n=8) embryos in the presence of GDNF. (**k, l**) Compiled representation of the total area of outgrowth from *Wnt1Cre/control* (black dots; n=15) and *Wnt1Cre/Ret* mutant (red dots; n=8) embryos in the presence of GDNF. See *Figure 5—figure supplement 1b* for rostrocaudal orientation of the midgut slices. The proximal midgut images shown in a, c, e, g represent sections that are 200–400 µm rostral to the

*Figure 6 continued on next page*

*Figure 6 continued*

apex of the midgut loop; the distal midgut images shown in b, d, f, h represent sections that are 200–400 μm caudal to the apex of the midgut loop (as indicated by the red boxes in i–l). p-Values: *p<0.05, **p<0.01, ***p<0.001, ns = not significant.

The online version of this article includes the following figure supplement(s) for figure 6:

**Figure supplement 1.** EDN3 exhibits minimal effect on E11.5 enteric progenitor outgrowth in vitro.

control Wnt1Cre-labeled cells (***Figure 7q***), also in the cytoplasm and nucleus (***Figure 7k***, magenta). However, phospho-Y1015 signal was only evident in 33% of *Ednrb*-deficient Wnt1Cre-derived cells (***Figure 7l and q***). Thus, a substantial subpopulation of the Wnt1Cre lineage requires Ednrb for Ret Y1015 phosphorylation. Because Ednrb is expressed only in a subset of Wnt1Cre-derived enteric progenitor cells (***Figure 7—figure supplement 1***), the residual Y1015 phosphorylation observed in *Wnt1Cre/Ednrb* mutant cells is likely to occur in the Ednrb-negative Wnt1Cre-derived cell population. Interestingly, *Ednrb* deficiency did not affect phosphorylation of Ret-Y1096 as demonstrated by positive staining in the Wnt1Cre+; Ret+ nuclei (***Figure 7n and r***). Furthermore, phospho-Y1096 staining was also notable within the Wnt1Cre-unlabled cell population (nGFP-negative, presumptively of the Pax2Cre lineage), although in these cells, both Ret and phospho-Y1096 Ret showed cell surface rather than nuclear localization (***Figure 7o and p***), suggesting that Ret signaling may be deployed differently in the Pax2Cre lineage. Although we (or other studies) have not yet proven the biological importance of Ret Y1015 in enteric progenitor migration, our results demonstrate that at least one feature of active Ret signaling in Wnt1Cre lineage cells requires the presence of Ednrb.

## Discussion

Using the Wnt1Cre and Pax2Cre lines, in this study we have deconstructed the involvement of the two major signaling axes (GDNF/Ret and Edn3/Ednrb) in ENS development. We have provided new insights on the mechanistic convergence between these signals and on the etiology of Hirschsprung disease.

The ENS has long been thought to derive solely from the neural crest. Our study reveals that the ENS has a dual lineage origin as defined by the Pax2Cre and Wnt1Cre drivers. It is important to clarify that we define these lineages by function (Cre recombination profile) rather than by anatomical origin. It is tempting to assume that these correspond strictly to placode and neural crest, respectively, as they do in craniofacial development (***Tan et al., 2023***). Nonetheless, for ENS progenitors this is not proven. Thus for example, it is formally possible that these recombination territories for ENS progenitors include different subdomains of neural crest rather than neural crest vs. placode, although even if so this would still fit the definition of dual lineages. For this reason we described the 'Pax2Cre lineage' and the 'Wnt1Cre lineage' in this manuscript rather than definitively claim these as placode and neural crest. Regardless, we present a compelling argument that these represent at least mostly distinct sublineages of the ENS: their gene expression profiles differ in early progenitor migration, their utilization of GDNF/Ret and Edn3/Ednrb signaling is distinct, the E12.5 consequences of Ret deficiency in the two lineages are radically distinct, and most importantly at least some of their terminal differentiation fates in the hindgut are unique. These latter two observations also preclude the possibility that Pax2Cre becomes active in migratory Wnt1Cre-derived cells. It is implausible that even a moderate percentage of the Pax2Cre lineage overlaps with the Wnt1Cre lineages: there is no contribution of the Pax2Cre lineage to numerous neural and nonneural structures that receive pre- or post-otic Wnt1Cre-labeled neural crest cells, and in the mature P9 colon, together the two lineages can account for all Hu+ neurons (***Figure 3—figure supplement 1g***), whereas any appreciable overlap would necessitate a third source of neurons. Our observations support reevaluation of the longstanding dogma of an exclusive neural crest origin of the ENS.

We observed that cells of the Pax2Cre and Wnt1Cre lineages co-mingle in the pharyngeal region and co-migrate through the gut. Indeed our observations indicate that gut migration by each lineage is dependent on the other. We cannot yet explain this behavior. Neural crest cells are known to migrate as an interconnected network, a property called collective cell migration (***Theveneau and Mayor, 2012***), and it is possible that Pax2Cre-derived cells incorporate into the same. The approx. 15% of cells of both lineages that do not express Ret (***Figure 2v***, gray bars) may also use this mechanism to support their migration. Within this assembly, the different roles of each lineage are unknown. Possible

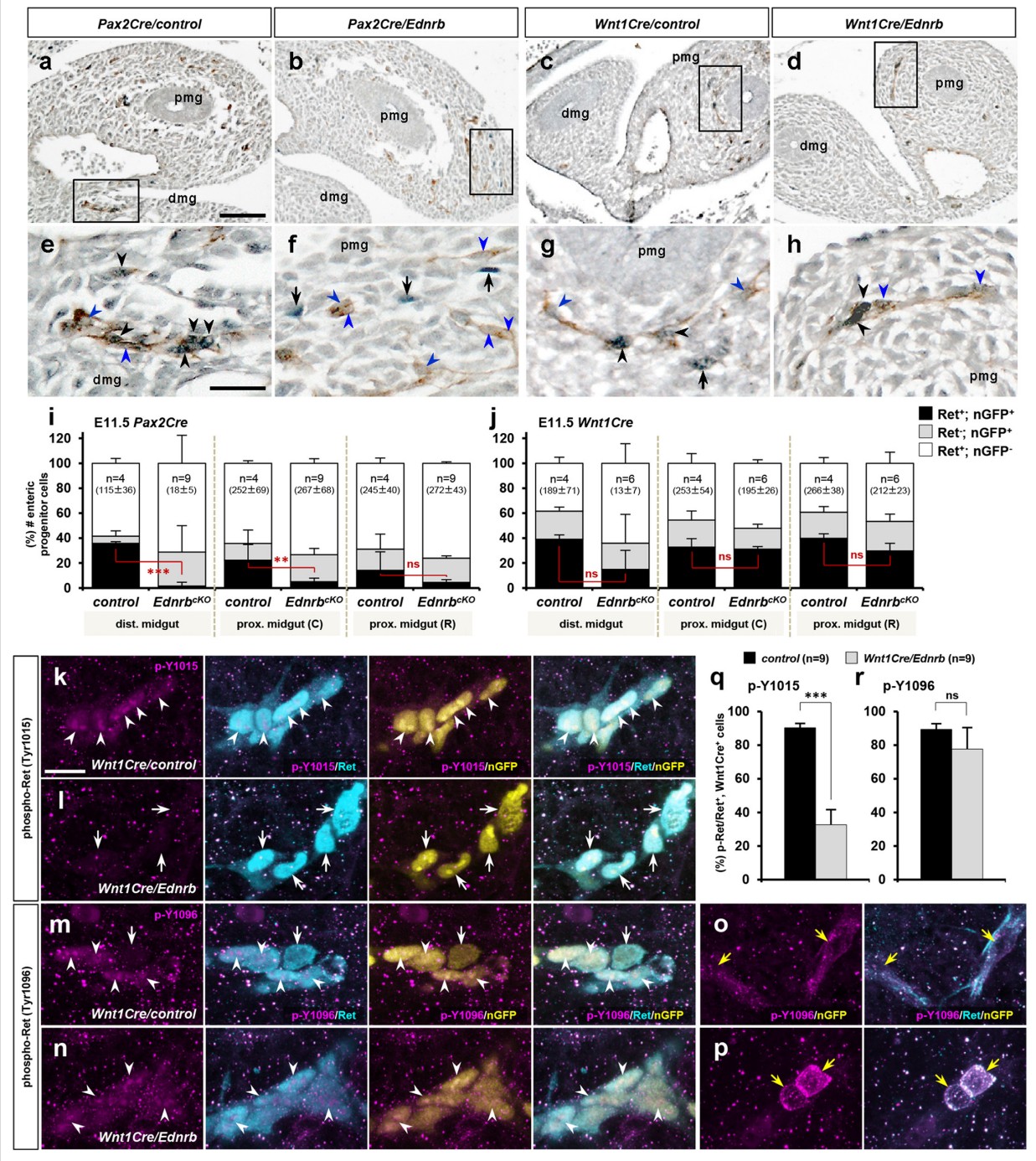

**Figure 7.** Lineage selective endothelin-dependent regulation of Ret in migratory enteric nervous system (ENS) progenitor cells. (**a–d**) Histological sections showing the apex of the midgut loop of E11.5 *Pax2Cre/Ednrb/Rosa26^{nT-nG}* mutant (**b**), *Wnt1Cre/Ednrb/Rosa26^{nT-nG}* mutant (**d**), and their littermate control embryos (a and c, respectively) stained for nuclear-GFP (blue) and Ret (brown). Magnified views of the bracketed areas (migratory wavefronts) in a–d are shown in e–h, respectively. Abbreviations: dmg, distal midgut; pmg, proximal midgut. (**i, j**) Compiled representations of the fractions of Ret^+, nGFP^+ (black); Ret^-, nGFP^+ (gray), and Ret^+, nGFP^- (white) ENS progenitor cells in the indicated areas of the embryos of the indicated genotype; (C) and (R) indicate caudal and rostral halves of the proximal midgut. The average total number of countable cells (labeled by either or both markers) in each gut segment is indicated in the parentheses. Error bars; mean ± s.d. (**k–p**) Confocal z-stack images of wavefront cells in wholemount preparations of E11.5 guts isolated from *Wnt1Cre/Ednrb/Rosa26^{nT-nG}* mutants (**l, n, p**) and their littermate control embryos (**k, m, o**) stained for nuclear-GFP (yellow), Ret (cyan), and phospho-Ret (Tyr1015) (magenta; **k, l**) or phospho-Ret(Tyr1096) (magenta; **m–p**). White arrowheads denote the presence of phosphorylated Ret in the Wnt1Cre-labeled cells, white arrows point to the absence of phosphorylated Ret in the Wnt1Cre-labeled cells, and yellow arrows denote cytoplasmic localization of phosphorylated Ret in nonlineage Ret^+ cells. (**q, r**) Compiled representations of phospho-Ret-(Tyr1015)^+

*Figure 7 continued on next page*

Figure 7 continued

fractions (**q**) and phospho-Ret-(Tyr1096)$^+$ fractions (**r**) at the wavefront (apex of the midgut loop) in *Wnt1Cre/Ednrb* mutants (n=9; gray bars) and their littermate control embryos (n=9; black bars). Error bars; mean ± s.e.m. p-Values: *p<0.05, **p<0.01, ***p<0.001, ns = not significant. Scale bars, 50 μm (**a–d**), 20 μm (**e–h, k–p**).

The online version of this article includes the following source data and figure supplement(s) for figure 7:

**Source data 1.** Raw data for panels i and j (numbers and fractions of Ret+ lineage-labeled, Ret- lineage -labeled, and Ret+ lineage-unlabeled cells in the distal midgut, caudal half of the proximal midgut and rostral half of the proximal midgut segments) and for panels q and r (numbers and fractions of pY1015+ or pY1096+ in Wnt1Cre-derived ENS progenitors at the migratory wavefront).

**Figure supplement 1.** Ednrb is expressed in subsets of Pax2Cre- and Wnt1Cre-derived enteric progenitor cells.

explanations for their required co-dependence could include unique abilities of each to attach to or degrade different components of the matrix, or some sort of paracrine activity that maintains collective migration. The net effect in normal development would be to bring both sets of progenitor cells to the same points of the gut at the same time, which might facilitate terminal assembly of circuitry. This observation of co-dependent migration might explain in part why the role of the Pax2Cre lineage was never previously recognized: were a subcomponent of the ENS able to migrate normally in neural crest ablation studies, the phenotypic result would likely not be described as aganglionosis.

Defecation is a function that is selectively accomplished by the myenteric plexus of the distal colon, where the integrated circuitry of mechanosensory neurons, excitatory and inhibitory interneurons, and excitatory and inhibitory motor neurons causes what are called 'giant migrating contractions' that push fecal material forward (*Sarna, 2010*). This is a unique neurophysiological process of the distal colon and is distinct from the spontaneous rhythmic contractions that occur with the passage of chyme (partially digested food) along the earlier axis of the gastrointestinal tract. In postnatal mice of each conditional *Ednrb* mutant background, in the proximal colon where cells of both lineages successfully migrate, we observed a selective loss of CGRP mechanosensory neurons in the *Pax2Cre/ Ednrb* background, and a selective loss of NOS$^+$ inhibitory motor neurons in the *Wnt1Cre/Ednrb* background. This observation is informative for demonstrating the developmental potential of each lineage, and corresponds to the fate of each lineage in normal mice, but is representative of only a very specific domain of the mutant guts. The explanation for Hirschsprung disease in both conditional *Ednrb* mutants is the total deficiency of both lineages in the distal colon, combined with the selective loss of specific neuronal subtypes in the transition zones (*Figure 3—figure supplement 3*) immediately prior to the aganglionic areas.

Using *Ret* conditional knockout, we observed that the Wnt1Cre and Pax2Cre lineages differ significantly in their use of GDNF signaling. Thus, the Wnt1Cre lineage requires GDNF responsiveness to migrate past the stomach, whereas the Pax2Cre lineage migrated into the midgut and only became GDNF-dependent at the hindgut. This behavior corresponds to expression of Ret in the early stages of Wnt1Cre-derived cells, and later abundance of Ret expression in Pax2Cre-derived cells. Alternative factors present in the foregut may uniquely support migration of the Pax2Cre lineage, but other explanations are possible.

Our studies demonstrate that Edn3 signaling through Ednrb is required to support GDNF responsiveness and thereby to support cell migration in both the Pax2Cre and Wnt1Cre lineages. We describe independent mechanisms to explain this relationship: Ret expression is induced or expanded by Ednrb signaling in Pax2Cre-derived cells, whereas in Wnt1Cre-derived cells, Ednrb induces competence for Ret signaling without changing Ret expression. It should be clear that we have not yet proven that either role is functionally required for normal ENS development, so for the present these are still candidate mechanisms. Regarding the latter, we determined that phosphorylation of Ret Y1015 requires Ednrb activity in Wnt1Cre-derived cells. Phosphorylation of this residue is known to activate the PLCg/PKC pathway (*Mahato and Sidorova, 2020*). For practical reasons, in this study we only investigated phospho-Y1015 and phospho-Y1096, and other modification sites or types might be more relevant to explain the relation between Ednrb and Ret in enteric progenitors of the Wnt1Cre lineage. In any event, the conceptual conclusion that Edn3-Ednrb signaling converges on GDNF-Ret function in both the Pax2Cre and Wnt1Cre lineages is supported by both the in vitro and in vivo observations and helps to explain the common phenotype of Hirschsprung disease that is seen when any component is mutated.

Our analyses have delineated the dual lineage origins of CGRP+ mechanosensory and NOS+ inhibitory motor neurons of the distal colonic circuitry, but clearly there is vastly more complexity in the ENS that is yet to be understood. Myenteric ganglia contain additional neural and glial subtypes that are still uncharacterized for lineage origin and function. Furthermore, the small intestine and proximal colon have their own physiology and functions, and it is expected that the ENS sustains the autonomy of these components of the gut through additional specialized circuits that respond to appropriate stimuli and execute different commands from those in the distal colon. As Wnt1Cre-derived and Pax2Cre-derived cells both colonize the foregut and midgut as well as the hindgut, we speculate that each lineage may also be allocated to distinct fates and functions in order to complete a functional circuitry in these parts of the gut.

We were able to define the distinct contributions of the Wnt1Cre and Pax2Cre lineages to the ENS using appropriate genetic strategies in mice. Cell-based transplantation approaches to restore bowel function in Hirschsprung disease infants are currently focused only on neural crest cells, e.g., as derived by directed differentiation from pluripotent stem cells. To fully reconstitute a functional ENS circuitry in patients, it will be necessary to ensure that these transplanted cells have the capacity to differentiate into the full range of ENS cell types, including mechanosensory neurons.

With the same null mutation of *Ednrb*, the severity of Hirschsprung disease is highly dependent on strain background in rats (*Dang et al., 2011*) and mice (*Hosoda et al., 1994*). This indicates the existence of unlinked alleles that modify the consequences of *Ednrb* deficiency. Similarly, in humans, Hirschsprung disease-associated mutations are almost always heterozygous (presumably haploinsufficient), but show variable extent of phenotypic penetrance, even so far as to be inherited in some affected patients from one asymptomatic parent (*Fuchs et al., 2001*). In some cases, the genetic background effect involves a mutation in a gene already known to be associated with Hirschsprung disease: a human genome-wide association study demonstrated genetic interaction between the *EDNRB* and *RET* loci (*Carrasquillo et al., 2002*), and synergy was evident in mice with intentional combinations of *Ednrb* and *Ret* (*McCallion et al., 2003*) or *Ednrb* and *Sox10* (*Cantrell et al., 2004*) alleles. Clearly there is opportunity to discover new genes that do not independently cause Hirschsprung disease but which modify the penetrance of the known set of genes. In this light, it was striking that mice on an ICR background with conditional or global *Ednrb* deficiency die consistently around 3 weeks of age. ICR mice are an outbred background with a genetic mixture of many poorly characterized ancestral sources (*Aldinger et al., 2009*) and have been maintained as an internally crossed (but not inbred) breeding colony by many providers. Interestingly, in congenital sensorineural deafness, the other major EDN3-EDNRB phenotype that with Hirschsprung disease defines Waardenburg-Shah syndrome type IV, the penetrance of hearing impairment in ICR background Edn3 or *Ednrb* mutant mice is variable (approx. 50%) (*Tan et al., 2023*). This indicates that different modifiers interact with Edn3-Ednrb in auditory development vs. ENS development. These mouse models may help to clarify the additional components and downstream pathways that result in human Hirschsprung and Waardenburg-Shah pathologies.

## Materials and methods

### Animals

*Ednrb* (*Rattner et al., 2013*) (JAX:011080), *Edn3* (*Baynash et al., 1994*) (JAX:002516), *Ret* (*Luo et al., 2007*) (JAX:028548), *Pax2Cre* (*Ohyama and Groves, 2004*), *Wnt1Cre* (*Danielian et al., 1998*), *ROSA26^{lacZ}* (*Soriano, 1999*), *ROSA26^{CAG-td-Tomato}* (*Madisen et al., 2010*) (JAX: 007914), *ROSA26^{nT-nG}* (*Prigge et al., 2013*) (JAX: 023537), *ROSA26^{EYFP}* (*Srinivas et al., 2001*) (JAX:006148), *Ednrb-EGFP* (*Gong et al., 2007*) (MMRRC_010620-UCD) alleles have been described previously. Wild-type ICR mice used for propagation of these lines were obtained from Harlan/Envigo. All alleles used in this study have been backcrossed for numerous generations to the ICR background, although because of intercrosses this was not rigorously controlled or quantified. All experiments with animals complied with National Institutes of Health guidelines and were reviewed and approved by the Medical University of South Carolina Institutional Animal Care and Use Committee (protocol #: 2018-00627).

## 24-hr fecal collection

2.5-Week-old animals were individually housed in a single mouse metabolic cage (Tecniplast) for 24 hr. Feces collected in a collection funnel were dried and measured for their dry weight.

## Postnatal distal colon preparation

Postnatal day 9 mice were anesthetized and transcardially perfused with PBS, and then the entire colon (including proximal colon) was dissected and transferred to PBS. After removing fecal materials by flushing PBS through the colon, a glass capillary tube was inserted to straighten the colon, and the colon was then fixed in an appropriate fixative at 4°C overnight. Fixatives used in this study were glutaraldehyde (0.2% glutaraldehyde, 0.4% paraformaldehyde, 2 mM $MgCl_2$ in PBS) for Xgal staining, and paraformaldehyde (4% PFA in PBS) for immunofluorescence staining.

## Wholemount Xgal staining

Fixed embryos, embryonic gut tissues, and postnatal colon preparations were washed with detergent rinse (0.02% NP40, 0.01% sodium deoxycholate, 2 mM $MgCl_2$ in PBS), and stained with Xgal staining solution (1 mg/ml Xgal in 0.1 M Tris-HCl pH = 7.4, 2 mM $MgCl_2$, 0.02% NP40, 0.01% sodium deoxycholate, 5 mM $_{K3}[Fe(CN)_6]$, 5 mM $_{K4}[Fe(CN)_6]$ in PBS) at room temperature overnight. The colon was opened by longitudinal incision, the (inner) submucosal and mucosal layers were removed, and the outer tissue (containing circular muscle, the myenteric plexus, longitudinal muscle, and serosa) was stained. Xgal-stained tissues were washed with PBS, and either or then used for in situ hybridization or wholemount immunostaining.

## In situ hybridization

Xgal-stained embryonic gut tissues were washed with PBST (1% Tween-20 in PBS), dehydrated in a methanol series ultimately to 100% methanol, then bleached in 3% $H_2O_2$ in methanol for 20 min at room temperature. Tissues were rehydrated in PBST, digested with 10 µg/ml Proteinase K in PBST for 4 min at room temperature, and then washed in 2 mg/ml glycine containing PBST. After post-fixation with 4% PFA for 10 min at room temperature, tissues were washed with PBST, equilibrated with hybridization solution (50% formamide, 5xSSC pH = 4.5, 1% SDS, 50 µg/ml yeast tRNA, 50 µg/ml heparin), and incubated with biotin-labeled RNA probe at 70°C overnight. Tissues were then washed with hybridization solution (without yeast tRNA and heparin) three times at 70°C, three times with high stringent wash (50% formamide, 2xSSC pH = 4.5) at 65°C, and washed with PBST at room temperature. Biotinylated probe was then detected by Streptavidin-HRP antibody followed by DAB chromogen reaction (see Wholemount immunostaining below).

Frozen sections were prepared from fixed embryos that were cryoprotected in 30% sucrose in PBS followed by embedding in OCT compound. Sections were post-fixed for 10 min in 4% PFA/PBS, washed with PBST, and then acetylated in 0.25% acetic anhydride containing 0.1 M triethanolamine for 5 min at room temperature. Section were equilibrated with hybridization solution (50% formamide, 5xSSC pH = 4.5, 50 µg/ml yeast tRNA), and incubated with DIG-labeled RNA probe at 58°C overnight. Sections were then washed with hybridization solution (without yeast tRNA) at 58°C, 2xSSC at room temperature, 2xSSC at 65°C, and followed by 0.1xSSC at 65°C. After wash with TBST (0.1% Tween-20 in TBS), DIG-labeled probe was detected by alkaline phosphatase-conjugated DIG antibody followed by NBT/BCIP reaction in NTMT (0.1 M NaCl, 0.1 M Tris-HCl pH = 9.5, 50 mM $MgCl_2$, 0.1% Tween-20) with 2 mM levamisole. In situ labeled sections were then subjected to immunofluorescence staining (see Immunofluorescence staining below). Riboprobes for *Edn3* and *Ednrb* genes are described previously (*Makita et al., 2008*).

## Two-color immunohistochemistry

5 µm continuous paraffin sections prepared from the head to tail of E10.5–12.5 *Pax2Cre/Ret/Rosa26^{nT-nG}*, *Pax2Cre/Ednrb/Rosa26^{nT-nG}*, *Wnt1Cre/Ret/Rosa26^{nT-nG}*, *Wnt1Cre/Ednrb/Rosa26^{nT-nG}*, and their littermate control embryos were deparaffinized in xylene, rehydrated through an ethanol sequence, and then microwaved for 25 min in citrate buffer (10 mM sodium citrate pH = 6.0) for antigen retrieval. Histological slides were bleached in 3% $H_2O_2$ in methanol for 2 min, washed in PBST (0.1% Tween-20 in PBS), and then first incubated with chicken anti-GFP antibody (1:3000, Abcam ab13970) at 4°C overnight. After three washes with PBST, sections were incubated with HRP-conjugated anti-chicken-IgY

(1:200, Jackson ImmunoResearch 703-035-155) for 2–3 hr at room temperature. GFP signal was then detected by 0.5 mg/ml DAB-blue solution containing 0.05% $CoCl_2$, 0.05% $Ni(NH_4)_2(SO_4)_2$ (pH = 7.2) with 0.015% $H_2O_2$. Nuclear-GFP immunolabeled sections were then bleached in 3% $H_2O_2$ in methanol for 2 min, washed in PBST (0.1% Tween-20 in PBS), and incubated with goat anti-Ret antibody (1:300, R&D Systems AF482) at 4°C overnight. After incubating with HRP-conjugated goat-IgG secondary antibody, Ret signal was detected using DAB-brown solution (0.5 mg/ml DAB with 0.03% $H_2O_2$). Slides were washed in $ddH_2O$, dehydrated in an ethanol sequence followed by xylene and mounted with DPX medium.

All sections containing foregut through midgut areas containing immunolabeled cells were captured as digital images using a Nikon DS-Fi1 digital camera in continuous order. For E10.5 analysis (*Figure 2v–x*), nGFP-positive, Ret-positive, and nGFP and Ret-double positive cells in the midgut segments were counted in every third section (15 µm intervals to avoid counting the same cell twice) using Fiji ImageJ plugin Cell Counter. For E11.5 analysis, nGFP-positive, Ret-positive, and nGFP and Ret-double positive cells in the rostral and caudal halves of the proximal midgut and in the distal midgut segments were counted separately in every third section (15 µm intervals to avoid counting the same cell twice), and compiled as whole midgut cell counts for lineage-specific Ret mutants (*Figure 2y and z*) or as cell counts of each segment for lineage-specific Ednrb mutants (*Figure 7i and j*).

## Embryonic gut slice preparation for collagen gel explant culture

Tomato⁺ embryos from *Pax2Cre/Ret/Rosa26^td-Tomato*, *Pax2Cre/Ednrb/Rosa26^td-Tomato*, *Wnt1Cre/Ret/Rosa26^td-Tomato*, *Wnt1Cre/Ednrb/Rosa26^td-Tomato,* and their littermate control embryos were collected in PBS under a fluorescence dissection scope (Nikon SMZ800), and midgut segments were dissected in PBS as described in *Figure 5—figure supplement 1*. For E10.5 explants, dissected midgut segments were embedded in 6% low melt agarose in PBS, and sectioned transversely from the caudal to rostral direction using a Vibratome 1000 microtome. Each 200 µm section was placed in a culture medium droplet on Petri dish and these were maintained in their caudal to rostral order until explanted in collagen gel. For E11.5 explants, both proximal and distal midgut segments were separately embedded in 6% low melt agarose in PBS, sectioned in a caudal to rostral and rostral to caudal direction, respectively, and placed in culture media droplets prior to explanting in collagen gel. To ensure similar stages of development between different litters, E10.5 tissue sections were only used if the midgut was 1.0–1.4 mm, and for E11.5 tissue if the proximal midgut segment was 1.6–2.0 mm.

## Collagen gel explant culture

Rat tail collagen and collagen gel preparation were performed as previously described (*Guthrie and Lumsden, 1994*) with minor modifications. Briefly, rat tail collagen was prepared by dissolving 2.5 mg/ml rat tail tendons in 0.5 N acetic acid at 4°C for 2 days. The solution was then centrifuged at 12,000 rpm for 1 hr at 4°C to remove debris, and the supernatant dialyzed against 0.1X Basal Eagle's Medium (BME) for 3 days at 4°C. For collagen gel explant culture, 4XBME and 7.5% sodium bicarbonate (with or without growth factor, see below) were gently placed into dialyzed rat tail collagen solution on ice to a final concentration of 1X and 0.2%, respectively. After very gentle pipetting up and down on ice, 25 µl collagen drops were placed onto a Petri dish, and then flattened by gently dropping the dish onto a flat surface several times. The collagen beds were then solidified in a tissue culture incubator (5% $CO_2$, 37°C) for 10 min. Once the collagen beds solidified into a gel, each embryonic gut slice was placed onto a collagen bed and mounted with 40 µl of freshly prepared collagen solution (1XBME, 0.2% $NaHCO_3$, with or without growth factor in dialyzed collagen), and the dish returned to the incubator for 10 min. Once the top collagen was fully solidified, culture medium with or without growth factor was added to the Petri dish to fully submerge the collagen gel domes. Gut slice explants were then cultured in 5% $CO_2$ at 37°C for 3 days. Culture media used was DMEM supplemented with 10% FBS, 100 U/ml penicillin/streptomycin, 100 µM putrescine (Sigma-Aldrich P5780), 10 ng/ml progesterone (Sigma-Aldrich P7556), 200 ng/ml selenious acid (Sigma-Aldrich 211176), 500 µM sodium butyrate (Sigma-Aldrich B5887), 1 mM sodium pyruvate (Sigma-Aldrich P2256), 5 µg/ml transferrin (Sigma-Aldrich T8158), 2 ng/ml triiodo-L-thyronine (T3) (Sigma-Aldrich T6397). Growth factors used were recombinant mouse GDNF (10 ng/ml, Abcam ab 56286), endothelin-3 (10 nM, Enzo LifeSciences ALX-155-003), and recombinant mouse NGF (10 ng/ml, Sino Biological 50385-MNAC). This EDN3 concentration promotes neurite outgrowth from

embryonic sympathetic and CNX ganglion explants (*Makita et al., 2008*) and NGF at the utilized concentration is widely used to promote growth and survival of variety of neuronal types (*Kuruvilla et al., 2004*). After 3 days, the Petri dishes containing the collagen gel domes were fixed in 4% PFA for 2 hr at 4°C and washed with PBS. The explanted tissue morphology and its Tomato$^+$ cell profiles were recorded in continuous axial order under bright field and epifluorescence settings, respectively (Nikon Eclipse 80i upright microscope). The collagen domes were then gently peeled off from the Petri dish and wholemount immunofluorescence-stained (see Immunofluorescence staining below) for confocal imaging. Total Tomato$^+$ outgrowth area was measured from epifluorescence images acquired from a Nikon Eclipse 80i upright microscope using Fiji ImageJ software; binary threshold selection of the Tomato$^+$ area of the original explanted tissue was subtracted from the terminal Tomato$^+$ area. For both E10.5 and E11.5 stages, the outgrowths of six sections (0–1200 µm range, see *Figure 5—figure supplement 1*) from each midgut segments were compiled into the dot plots. If individual explanted slices failed to stay in the collagen gel their individual data were not included in the analysis.

## Wholemount immunostaining

Fixed tissues (mouse embryos, embryonic guts, postnatal gut slices, postnatal distal colon preparations) were dehydrated in a methanol series (to 100%), and bleached overnight in 3% $H_2O_2$ in 20% DMSO containing methanol solution. Tissues were then rehydrated in PBST (1% Tween-20 in PBST), and incubated in blocking solution (1% non-fat dried milk, 20% DMSO in PBST) for 10–60 min at room temperature followed by incubation with primary antibody in blocking solution for 2–3 days at 4°C. After extensive washes with PBST, tissues were incubated with HRP-conjugated secondary antibody (1:200, Jackson ImmunoResearch) for 2–3 days at 4°C. Immunoreactive signal was then visualized by DAB detection (0.2 mg/ml DAB in PBST with 0.03% $H_2O_2$). Stained tissues were cleared with Sca*l*eU2 (4 M urea, 30% glycerol, 0.1% Triton X-100) for imaging. Antibodies used included rabbit anti-Pax2 (1:50, Invitrogen 08-1483), rabbit anti-AP2 (1:1000, Abcam ab52222), goat anti-Sox10 (1:200, Santa Cruz sc-17342), mouse anti-HuD (1:500, Santa Cruz sc-13577), mouse anti-Tuj1 (1:1000, BioLegend MMS-435P), rabbit anti-CGRP (1:500-1000, Millipore PC205L), and rabbit anti-NOS1 (1:500, Santa Cruz sc-648). Stained tissues were cleared with Sca*l*eU2 and flat-mounted to histology slides for imaging. CGRP$^+$ afferent projections in colonic mucosa were quantified by binary threshold-level selection of the CGRP$^+$ pixel area per single villus and NOS$^+$ efferent projections were also quantified by binary threshold-level selection of NOS$^+$ pixel area in the circular muscle layer, both using Fiji-ImageJ software.

## Wholemount immunofluorescence staining

Fixed tissues (embryonic vibratome sections, embryonic gut tissues, collagen gel cultured explants, postnatal colon vibratome sections, and preparations) were permeabilized in PBST (1% Tween-20 in PBS), incubated in blocking solution (1% non-fat dried milk, 20% DMSO in PBST) for 10–60 min at room temperature followed by incubation with primary antibody in blocking solution for 2–3 days at 4°C. After extensive washing with PBST, tissues were incubated with Alexa Fluor-conjugated secondary antibody (1:500, Invitrogen) for 2–3 days at 4°C. Immunolabeled tissues were counterstained with DAPI and cleared with Sca*l*eU2 and mounted to histology slides for confocal imaging. Antibodies used were rabbit anti-Pax2 (1:50, Invitrogen 08-1483), rabbit anti-AP2 (1:1000, Abcam ab52222), goat anti-Sox10 (1:200, Santa Cruz sc-17342), mouse anti-2H3 (1:100, DSHB), chicken anti-GFP (1:3000, Abcam ab13970), goat anti-p75 (1:300, R&D Systems AF1157), mouse anti-HuD (1:500, Santa Cruz sc-13577), rabbit anti-BLBP (1:500, Abcam ab32423), mouse anti-Tuj1 (1:1000, BioLegend MMS-435P), rabbit anti-CGRP (1:500-1000, Millipore PC205L), rabbit anti-NOS1 (1:500, Santa Cruz sc-648), goat anti-Ret (1:300, R&D Systems AF482), rabbit anti-phospho-Ret (Tyr1015) (1:300, Themo Fisher PA5-105930), and rabbit anti-phospho-Ret (Tyr1096) (1:300, Themo Fisher PA5-105796). Fiji ImageJ plugin cell counter was used for counting Tomato$^+$, CGRP$^+$, NOS$^+$, GFP$^+$, and Hu$^+$ ganglion cells in myenteric plexuses of the distal colon (defined on the basis of perfusion by the inferior mesenteric artery), and for counting Ret$^+$, phospho-Ret Tyr1015$^+$, phospho-Ret Tyr1096$^+$, and GFP$^+$ cells in E11.5 embryonic ENS wavefront cells.

## Statistics

All quantified data were graphed as mean ± s.e.m. or mean ± s.d. (as indicated in the figure legend), and analyzed for significance using a two-tailed Student's t-test. Dot plots show the each individual data points and mean.

## Materials availability

This study did not generate new unique reagents.

## Additional information

### Funding

| Funder | Grant reference number | Author |
|---|---|---|
| National Institute of Neurological Disorders and Stroke | NS084121 | Takako Makita |

The funders had no role in study design, data collection and interpretation, or the decision to submit the work for publication.

### Author contributions

Denise M Poltavski, Alexander T Cunha, Jaime Tan, Investigation, Writing – original draft, Writing – review and editing; Henry M Sucov, Conceptualization, Writing – original draft, Writing – review and editing; Takako Makita, Conceptualization, Data curation, Formal analysis, Supervision, Investigation, Writing – original draft, Project administration, Writing – review and editing

### Author ORCIDs

Denise M Poltavski ⓘ https://orcid.org/0009-0002-7377-4915
Alexander T Cunha ⓘ https://orcid.org/0000-0002-7178-8152
Jaime Tan ⓘ https://orcid.org/0009-0008-3063-8060
Henry M Sucov ⓘ https://orcid.org/0000-0002-3792-3795
Takako Makita ⓘ https://orcid.org/0000-0002-5598-5690

### Ethics

All experiments with animals complied with National Institutes of Health guidelines and were reviewed and approved by the Medical University of South Carolina Institutional Animal Care and Use Committee (protocol #2018-00627).

Reviewer #1 (Public Review): https://doi.org/10.7554/eLife.96424.3.sa1
Reviewer #2 (Public review): https://doi.org/10.7554/eLife.96424.3.sa2
Author response https://doi.org/10.7554/eLife.96424.3.sa3

## Additional files

### Supplementary files
• MDAR checklist

### Data availability
All raw data are included as source data files.

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
