## [Editor Report · eLife Assessment]

This study provides **valuable** insights into our understanding of the development of the enteric nervous system. The authors use genetically engineered mice to study the behavior of stem cells in organizing the enteric nervous system and the secreted signals that regulate these cells. The study rests on a degree of **incomplete** evidence since the characterization of some of the mouse resources is not complete in the current version.

---

## [Referee Report · Reviewer #1 (Public Review)]

Summary:

The manuscript by Poltavski and colleagues describes the discovery of previously unreported enteric neural crest-derived cells (ENCDC) which are marked by Pax2 and originating from the Placodes. By creating multiple conditional mouse mutants, the authors demonstrate these cells are a distinct population from the previously reported ENCDCs which originate from the Vagal neural crest cells and express Wnt1.

These Pax2-positive ENCDCs are affected due to the loss of both Ret and Ednrb highlighting that these cells are also ultimately part of the canonical processes governing ENCDC and enteric nervous system (ENS) development. The authors also make explant cultures from the mouse GI tract to detect how Ednrb signaling is important for Ret signaling pathways in these cells and rediscovers the interactions between these 2 pathways. One important observation the authors make is that CGRP-positive neurons in the adult distal colon seem to be primarily derived from these Pax2-positive ENCDCs, which are significantly reduced in the Ednrb mutants, thus highlighting the role of Ednrb in maintaining this neuronal type.

Comments on latest version:

Author response: We disagree that the datasets from previous studies provide additional insights that are relevant to the current study. It must be appreciated that Wnt1Cre and Pax2Cre are genetic lineage tracers and that migratory ENS progenitor cells labeled with these reagents do not maintain expression of Wnt1 and Pax2 mRNA or protein. The Wnt1 and Pax2 genes are only transiently expressed within their distinct regions of the ectoderm, and their expression turns off as cells delaminate and begin migration. Thus, Pax2Cre-labeled ENS progenitor cells are not Pax2-positive thereafter. The single cell RNA-Seq studies suggested by the reviewer were collected from older embryos and postnatal mice, and do not represent the E10.5-E11.5 period that accounts for genesis of Ret-mediated and Ednrb-mediated Hirschsprung disease pathology. Even with the most recent work by Zhou et al (Dev Cell, 2024) that included E10.5 cells, this analysis only evaluated neural crest-derived Sox10Cre lineage cells, which does not include the placode-derived Pax2Cre lineage (as we show explicitly in Fig. 2-figure supplement 2). Consequently, it would not be possible to find the "Pax2-positive cells" in these datasets. Performing a new transcriptomic analysis by isolating Pax2Cre-lineage and Wnt1Cre-lineage cells at the appropriate developmental time points could be the basis of future studies, but we think these are beyond the scope of the present paper.

Reviewer comment: Since these cells are a completely new discovery, additional validation would be beneficial. Whole early GI tract datasets are available, such as human 6-week fetal gut data (PMID: 29802404) and whole mouse embryo studies spanning development that include ENS (PMID: 38355799). If the authors believe that none of these existing datasets can detect these cells in their developmental state and that targeted cell studies with specific Cre drivers would be required, they should make this explicitly clear.

A key advantage of discovering a new cell type, particularly in the relatively understudied field of ENS, is the opportunity for the broader community to leverage this finding to inform their own research. If these cells are absent from current datasets, even those covering the whole GI tract, this should be clearly communicated.

I aim to support the authors here. New discoveries in science require robust validation to enhance their impact. The authors have generated an important reagent with great potential for broader use, and addressing these straightforward requests would strengthen the study and make it more valuable to the scientific community.

Author response: The observation that human mutations in RET and EDNRB both cause Hirschsprung disease is decades old, and of course numerous studies in human, mouse, and cells have addressed the relation between the two signaling pathways. We did not mean to imply that we were the first to discover that Ret and Ednrb signaling pathways interact. The reviewer cites a number of papers all from the Chakravarti lab that address this phenomenon; while these are a valuable contribution to the field, there is still more to be learned. The model elaborated in PMID: 31313802, in which Ret and Ednrb are both enmeshed in a common gene regulatory network, does not readily explain why each has a different phenotypic manifestation and doesn't take into account the importance of the placodal lineage. The main new contributions of our paper are the existence of a new cell lineage that contributes to the ENS, and that the placodal and neural crest lineages utilize Ret and Ednrb signaling differently. The clarification of how these elements are differentially used by the two lineages explains long-segment and short-segment Hirschsprung disease (Ret and Ednrb mutants, respectively) far better than in past studies. The reviewer unfortunately dismisses these insights and seems to feel that a biochemical exploration of one specific component of the signaling interaction (Y1015 phosphorylation) would be more relevant. This should be the basis of future studies and are beyond the scope of the new findings reported in the present paper

Reviewer comment: The authors completely miss the point. There is no association between phenotypic severity (L-HSCR, S-HSCR, or TCA) and mutations in a given gene in HSCR. EDNRB, for example, has a syndromic association with Waardenburg-Shah syndrome (WS4-A), which includes pigmentation anomalies due to EDNRB expression in neural crest cells that give rise to pigment cells.

The authors' discovery reinforces the current paradigm that nearly all HSCR is mediated by mutations in genes within the GRN, accounting for 72% of the population attributable risk. This is valuable; reinforcing established paradigms with new data is crucial, and the authors should appreciate the significance of this contribution.

The discovery of the signaling interaction is particularly important, as it offers a potential explanation for disease severity and provides a basis for classifying patients in future sequencing studies. It is surprising that the authors seem reluctant to highlight this novel finding, as it could greatly benefit future research, including the development of specific mouse mutants and advancing human genetics studies.

Author response: The reviewer overlooked that one of the review articles that we cited (Chen, Hsu, & Hung, 2020) has a dedicated paragraph for RET (section 3.14), which summarizes the work by Barheri-Yarmand et al (PMID: 25795775) which is the very paper noted by the reviewer in the comment above. The reviewer also somewhat misstated the results of the Barheri-Yarmand et al study. By immunostaining, this paper showed nuclear localization of endogenous Ret, albeit a version of Ret with a disease-associated mutation that makes it constitutively active by constitutive autophosphorylation. Nonetheless, this was endogenous Ret. The paper also used overexpression of GFP-tagged RET in HEK293 cells to show that wildtype RET can behave in a similar manner, at least under these circumstances. Our point is simply that Ret (and other receptor tyrosine kinases) can be found in the nucleus in certain biological contexts, and our observations are consistent with this precedent. The reviewer also suggests a biochemical follow-up analysis related to this observation, which we agree would be of interest. Such an investigation however is beyond the scope of the present study.

Reviewer comment: As the authors themselves now highlight from the cited paper that any evidence of RET entering the nucleus is of a mutant RET protein, How does this align with their discovery for wildtype protein?

This finding of nuclear localization of RET is both intriguing and unprecedented. Despite extensive biochemical studies on RET, given its role as an oncogene, this feature has not been identified before. If validated, this discovery could significantly advance the field and improve interpretation of future studies. I reiterate my previous point: a novel finding that challenges the current paradigm requires additional evidence.

---

## [Referee Report · Reviewer #2 (Public review)]

Summary:

This manuscript by Poltavski and colleagues explores the relative contributions of Pax2- and Wnt1- lineage derived cells in the enteric nervous system (ENS) and how they are each affected by disruptions in Ret and Endrb signaling. The current understanding of ENS development in mice is that vagal neural crest progenitors derived from a Wnt1+ lineage migrate into and colonize the developing gut. The sacral neural crest was thought to make a small contribution to the hindgut in addition but recent work has questioned that contribution and shown that the ENS is entirely populated by vagal crest (PMID: 38452824). GDNF-Ret and Endothelin3-Ednrb signaling are both known to be essential for normal ENS development and loss of function mutations are associated with a congenital disorder called Hirschsprung's disease. The transcription factor Pax2 has been studied in CNS and cranial placode development but has not been previously implicated in ENS development. In this work, the authors begin with the unexpected observation that conditional knockout of Ednrb in Pax2-expressing cells causes a similar aganglionosis, growth retardation, and obstructed defecation as conditional knockout of Ednrb in Wnt1-expressing cells. The investigators then use the Pax2 and Wnt1 Cre transgenic lines to lineage-trace ENS derivatives and assess the effects of loss of Ret or Ednrb during embryonic development in these lineages. Finally, they use explants from the corresponding embryos to examine the effects of GDNF on progenitor outgrowth and differentiation.

Strengths:

- The manuscript is overall very well illustrated with high resolution images and figures. Extensive data are presented.

- The identification of Pax2 expression as a lineage marker that distinguishes a subset of cells in the ENS that may be distinct from cells derived from Wnt1+ progenitors is an interesting new observation that challenges current understanding of ENS development

- Pax2 has not been previously implicated in ENS development - this manuscript does not directly test that role but hints at the possibility

- Interrogation of two distinct signaling pathways involved in ENS development and their relative effects on the two purported lineages

Weaknesses:

- The major challenge with interpreting this work is the use of two transgenic lines, Wnt1-Cre and Pax2-Cre, which are not well characterized in terms of fidelity to native gene expression and recombination efficiency in the ENS. If 100% of cells that express Wnt1 do not express Cre or if the Pax2 transgene is expressed in cells that do not normally express Pax2, then these observations would have very different interpretations and would not support the conclusions made. The two lineages are never compared in the same embryo, which also makes it difficult to assess relative contributions and renders the evidence more circumstantial than definitive.

- Visualization of the Pax2-Cre and Wnt-1Cre induced recombination in cross-sections at postnatal ages would help with data interpretation. If there is recombination evident in the mesenchyme, this would particularly alter interpretation of Ednrb mutant experiments, since that pathway has been shown to alter gut mesenchyme and ECM, which could indirectly alter ENS colonization.

- The data on distinct lineages in Fig 3 is somewhat weak and the description in the Results section tends to over-interpretation. For example, "A minimum number (approx. 3%) of CGRP+ neurons were labeled by Wnt1Cre ... which indicates that Wnt1Cre-derived cells have little or no commitment to a mechanosensory fate in the distal colon." The data panel in Fig 3f shows that most of the CGRP-IR cells in Wnt1-Cre-Tomato mice are tdTomato+ though their tdTomato fluorescence is less intense than in neighboring smaller, likely glial cells. This suggests that CGRP+/Tomato+ neurons were likely undercounted. IHC for tdTomato to ensure detection of low levels of Tomato expression and quantification of observations would strengthen the authors' claim. CGRP+ enteric neurons have been visualized and functionally described by several investigators in the field using Wnt1-Cre-GCaMP mice, which also challenges the authors' conclusions. Finally, quantification of CGRP+ enteric neurons by measuring CGRP mucosal fiber immunoreactivity is not accurate because it would reflect both ENS CGRP-expressing neurons and visceral afferents from DRG. Moreover, it is not known if all CGRP+ enteric neurons project to the mucosa or if all mucosal-projecting neurons are mechanosensory. Finally, most of the signal seems to be non-specific background staining in the mucosa and quantification of mucosal signal in this context does not seem meaningful.

- No consideration of glia - are these derived from both lineages?

- No discussion of how these observations may fit in with recent work that suggests a mesenchymal contribution of enteric neurons (PMID: 38108810)

- Phospho-RET staining in Figure 7 is difficult to discern and interpret with high background. Positive and negative controls would strengthen these data.

Comments on revised version:

The authors have responded to the weaknesses identified above. Based on my own assessment of the revised manuscript, my assessment is unchanged because the manuscript is largely unchanged.

---

## [Author Response]

The following is the authors’ response to the original reviews.

**Reviewer #1 (Public Review):**
The manuscript by Poltavski and colleagues describes the discovery of previously unreported enteric neural crestderived cells (ENCDC) which are marked by Pax2 and originating from the Placodes. By creating multiple conditional mouse mutants, the authors demonstrate these cells are a distinct population from the previously reported ENCDCs which originate from the Vagal neural crest cells and express Wnt1.These Pax2-positive ENCDCs are affected due to the loss of both Ret and Ednrb highlighting that these cells are also ultimately part of the canonical processes governing ENCDC and enteric nervous system (ENS) development. The authors also make explant cultures from the mouse GI tract to detect how Ednrb signaling is important for Ret signaling pathways in these cells and rediscovers the interactions between these 2 pathways. One important observation the authors make is that CGRP-positive neurons in the adult distal colon seem to be primarily derived from these Pax2-positive ENCDCs, which are significantly reduced in the Ednrb mutants, thus highlighting the role of Ednrb in maintaining this neuronal type.I appreciate the amount of work the authors have put into generating the mouse models to detect these cells, but there isn't any new insight on either the nature of ENCDC development or the role of Ret and Ednrb. Also, there are sophisticated single-cell genomics methods to detect rare cell type/states these days and the authors should either employ some of those themselves in these mouse models or look at extensively publicly available single-cell datasets of the developing wildtype and mutant mouse and human ENS to map out the global transcriptional profile of these cells. A more detailed analysis of these Pax2-positive cells would be really helpful to both the ENS community as well as researchers studying gut motility disorders.

We would like to point out that the reviewer’s comments in both Public Review and in some cases reiterated in Recommendations for the Authors are rooted in several misunderstandings. The reviewer writes “Pax2-positive ENCDCs”, as if the Pax2 lineage (properly, the Pax2Cre-labeled lineage) of the ENS is a subset of neural crest, and states that “there isn’t any new insight” from our study on ENS development. Our conclusion is quite different, that the Pax2Cre lineage (placode-derived) is distinct from the neural crest-derived cell lineage. The reviewer may not have appreciated that our study establishes a fundamental reinterpretation of the very long-standing dogma that the ENS is derived solely from neural crest. We believe that finding and characterizing the unique contribution of an independent cell lineage to the ENS provides critical new perspectives into ENS development and the etiology of Hirschsprung disease. One feature of the Pax2Cre (placodal) lineage is as the source of CGRP-positive mechanosensory neurons in the colon (as the reviewer mentioned), but this is one feature of the larger conceptual discovery of the existence of a separate lineage contribution to the ENS, not the most important observation in and of itself.

The reviewer continues by saying that we “rediscovered” the interaction between Ednrb and Ret in ENS development. In our study we show that the two lineages (placode-derived and neural crest-derived) employ Ednrb and Ret signaling in distinct ways. This isn’t simply rediscovery, this is new insight. To the extent that both lineages utilize both signaling axes (albeit with mechanistic differences) is a primary reason why the unique placodal lineage contribution to the ENS remained unsuspected until now. We have revised the text to make these points more clear in our revised manuscript.

The reviewer also suggests single cell genomic methods, which is addressed below in our response to the reviewer’s first recommendation.

**Reviewer #2 (Public Review):**
This manuscript by Poltavski and colleagues explores the relative contributions of Pax2- and Wnt1- lineagederived cells in the enteric nervous system (ENS) and how they are each affected by disruptions in Ret and Endrb signaling. The current understanding of ENS development in mice is that vagal neural crest progenitors derived from a Wnt1+ lineage migrate into and colonize the developing gut. The sacral neural crest was thought to make a small contribution to the hindgut in addition but recent work has questioned that contribution and shown that the ENS is entirely populated by the vagal crest (PMID: 38452824). GDNF-Ret and Endothelin3-Ednrb signaling are both known to be essential for normal ENS development and loss of function mutations are associated with a congenital disorder called Hirschsprung's disease. The transcription factor Pax2 has been studied in CNS and cranial placode development but has not been previously implicated in ENS development. In this work, the authors begin with the unexpected observation that conditional knockout of Ednrb in Pax2-expressing cells causes a similar aganglionosis, growth retardation, and obstructed defecation as conditional knockout of Ednrb in Wnt1-expressing cells. The investigators then use the Pax2 and Wnt1 Cre transgenic lines to lineage-trace ENS derivatives and assess the effects of loss of Ret or Ednrb during embryonic development in these lineages. Finally, they use explants from the corresponding embryos to examine the effects of GDNF on progenitor outgrowth and differentiation.Strengths:- The manuscript is overall very well illustrated with high-resolution images and figures. Extensive data are presented.- The identification of Pax2 expression as a lineage marker that distinguishes a subset of cells in the ENS that may be distinct from cells derived from Wnt1+ progenitors is an interesting new observation that challenges the current understanding of ENS development.- Pax2 has not been previously implicated in ENS development - this manuscript does not directly test that role but hints at the possibility.- Interrogation of two distinct signaling pathways involved in ENS development and their relative effects on the two purported lineages.

The reviewer provided a succinct and accurate summary of our analysis. We correct just the one statement that the ENS is entirely populated by vagal crest. The paper cited by the reviewer (PMID: 38452824) used Wnt1DreERT2 to lineage label the NC population, so of course only looked at neural crest (comparing vagal vs. sacral NC). The advance in our study is to newly document the independent contribution of the placodal lineage.

Weaknesses:- The major challenge with interpreting this work is the use of two transgenic lines, rather than knock-ins, Wnt1Cre and Pax2-Cre, which are not well characterized in terms of fidelity to native gene expression and recombination efficiency in the ENS. If 100% of cells that express Wnt1 do not express this transgene or if the Pax2 transgene is expressed in cells that do not normally express Pax2, then these observations would have very different interpretations and not support the conclusions made. The two lineages are never compared in the same embryo, which also makes it difficult to assess relative contributions and renders the evidence more circumstantial than definitive.

We do not agree that the Cre lines being transgenics rather than knock-ins changes the utility of these reagents or the interpretation of the results; there are also potential problems with knock-in alleles. Wnt1Cre has been in use for 25 years as a pan-neural crest lineage cell marker with exceptional efficiency and specificity (including numerous studies of the ENS), so we disagree that it is not well characterized. Pax2Cre of course has not previously been studied in the ENS, but it has been broadly used in other contexts (e.g., craniofacial, kidney). That said, and as noted in our original manuscript, we are aware that an issue of this study is the uniqueness of the recombination domains of the two Cre lines. As we wrote, Wnt1Cre and Pax2Cre cannot be combined into the same embryo because they are both Cre lines, and we do not have a suitable nonCre recombinase line to substitute for either. Instead, we demonstrate that the two lines recombine in distinct territories of the early embryonic ectoderm, and that the two lineages thus labeled are distinct in marker expression at the initial onset of their delamination, utilize Edn3-Ednrb and GDNF-Ret in distinct ways during their migration to the hindgut, and contribute to different terminal cell fates in the colon. We think this evidence of the distinct nature of the two lineages from start to finish is compelling rather than merely circumstantial.

- Visualization of the Pax2-Cre and Wnt-1Cre induced recombination in cross-sections at postnatal ages would help with data interpretation. If there is recombination induced in the mesenchyme, this would particularly alter the interpretation of Ednrb mutant experiments, since that pathway has been shown to alter gut mesenchyme and ECM, which could indirectly alter ENS colonization.

We have several thoughts about this comment. First, we are uncertain why postnatal analysis would be informative, as ENS colonization occurs (or fails to occur in mutants) during embryogenesis. The reviewer might be thinking of a juvenile stage additional contribution to the ENS, which is addressed below (responses to *Recommendations for the Authors*) but as we discuss there is not relevant to our analysis. Second, we did examine recombination in the distal hindgut at E12.5 during ENS colonization (Fig. 1f and 1h) and did not see overlap between either Cre recombination domain and *Edn3* mRNA expression (which is expressed by the nonENS mesenchyme). Furthermore, Ednrb is not expressed in the gut mesenchyme during ENS colonization (Fig. 7figure supplement 1), thus ectopic mesenchymal Cre expression, if any, by either line would have no impact in Cre/Ednrb mutants. Lastly, the reviewer’s idea could have been a plausible hypothesis at the onset of the project, but here we show positive evidence for a different explanation. We do not rigorously exclude the reviewer’s hypothesis, nor other theoretically possible models, but we think we have provided a strong case to support the direct involvement of Ret and Ednrb in ENS progenitors rather than in surrounding non-neural mesenchyme.

- No consideration of glia - are these derived from both lineages?

To properly address this question would require new reagents and analyses that we have not yet initiated. While an interesting question from a developmental biology standpoint, we don’t think that this investigation would change any of the interpretations that we make in the manuscript.

- No discussion of how these observations may fit in with recent work that suggests a mesenchymal contribution of enteric neurons (PMID: 38108810).

The recent paper cited by the reviewer is very explicit in describing this mesenchymal contribution to the ENS as occurring after postnatal day P11. Other than the terminal Hirschsprung phenotype, all of our analysis of cell lineage migration and fate and colonic aganglionosis was conducted at embryonic or early (P9) postnatal stages. We therefore do not see a relation of our work to this study. In light of this paper, however, we do agree that it would be worthwhile in a future study to explore Wnt1Cre and Pax2Cre lineage dynamics in the ENS of older mice.

**Reviewer #1 (Recommendations For The Authors):**
(1) The authors should reanalyze multiple single-cell RNA-seq datasets available now, to see if these cells are detected in those studies and then look at the global transcriptional profile of these Pax2-positive cells compared to the other vagal neural crest-derived ENCDCs. Some of these datasets can be found here - PMIDs: 33288908, 37585461, and https://www.gutcellatlas.org/.

We disagree that the datasets from previous studies provide additional insights that are relevant to the current study. It must be appreciated that Wnt1Cre and Pax2Cre are genetic lineage tracers and that migratory ENS progenitor cells labeled with these reagents do not maintain expression of Wnt1 and Pax2 mRNA or protein. The *Wnt1* and *Pax2* genes are only transiently expressed within their distinct regions of the ectoderm, and their expression turns off as cells delaminate and begin migration. Thus, Pax2Cre-labeled ENS progenitor cells are not Pax2-positive thereafter. The single cell RNA-Seq studies suggested by the reviewer were collected from older embryos and postnatal mice, and do not represent the E10.5-E11.5 period that accounts for genesis of Ret-mediated and Ednrb-mediated Hirschsprung disease pathology. Even with the most recent work by Zhou et al (Dev Cell, 2024) that included E10.5 cells, this analysis only evaluated neural crest-derived Sox10Cre lineage cells, which does not include the placode-derived Pax2Cre lineage (as we show explicitly in Fig. 2-figure supplement 2). Consequently, it would not be possible to find the “Pax2-positive cells” in these datasets. Performing a new transcriptomic analysis by isolating Pax2Cre-lineage and Wnt1Cre-lineage cells at the appropriate developmental time points could be the basis of future studies, but we think these are beyond the scope of the present paper.

(2) Even in their current quantification method of using immunofluorescent cells in a microscopic field, the authors count very few cells. The quantification in Figures 2v-2z is only from 4 embryos and is in the hundreds. This leads to misrepresentation of cell numbers and is best reflected in Figure 2x, where Wnt1Cre/Ret GI tracts have 0 Ret +ve cells, which we now know is not true even in ubiquitous Ret null embryos, where Ret null cells are detected as late as E14.5 (PMID 37585461)

Because of the reviewer’s comment, we recognize that the specific detail about cell numbers wasn’t properly written. We didn’t count a few hundred cells total, it was a few hundred cells per embryo. Exact numbers are provided in the revised figure legend where “cells/embryo” is now explicitly stated. Multiplied by the number of embryos, this means that we evaluated approx. 1000 total cells per genotype and time point in cases where Ret+ and/or GFP+ (lineage+) cells were found. The total absence of such cells in Wnt1Cre/Ret mutants is a rigorous conclusion. Our results do not misrepresent nor contradict the study by Vincent et al (PMID 37585461). Our analyses were performed on gut tissue isolated at E10.5 and E11.5 stages, which is long before Schwann cell precursors (SCPs, the primary focus of the Vincent et al study) colonize the gut (E14.5; Uesaka et al, 2015. PMID: 26156989). Indeed, as the reviewer notes, SCPs migrate into the gut in a Retindependent manner. For being at a much earlier time point, our focus is on the cranial ectoderm sources of ENS progenitors. We have adjusted the text associated with Fig. 2 to make this more clear.

(3) There are multiple sections in the manuscript that rehash already known facts, like the whole section about Wnt1 conditional Ret null mice which show failure of migration of ENCDCs. This has been shown multiple times and doesn't add anything to the author's story.

We think this comment stems from the reviewer’s perception that the Pax2Cre lineage is a subset of neural crest. The Wnt1Cre data (including Ret-deficient and Ednrb-deficient embryos) presented in the manuscript are not intended to rehash what is already known but to establish important similarities and differences between the newly identified placode-derived and the well-established neural crest-derived ENS progenitor cells. In light of the reviewer’s suggestion #8 below, to move the Wnt1Cre lineage analysis to a supplement, this information remains in the main text to provide proper comparison to the Pax2Cre-lineage profile. We think we were fair in the text to the legacy of work on neural crest and ENS development and were explicit in using our Wnt1Cre analysis to compare to the Pax2Cre lineage. Finally, we point out that our analysis was conducted on a different genetic background (outbred ICR) compared to previous studies, and there are strain-specific differences in Hirschsprung-associated lethality between our background and previous studies, so it was not impossible that the behavior of the neural crest cell lineage in the ICR background could be different from past observations on different backgrounds. Although we did not identify any major differences, it is important that the information on NC behavior in this background be presented.

(4) Also, the conclusion drawn for Figure 5C "this indicates that the Wnt1Cre-derived cells do not harbor a cellautonomous response to GDNF" seems to suggest the authors are not very well versed with the ENS literature. GDNF as well as EDN3 are expressed from surrounding mesenchyme and are cell non-autonomous.

The reviewer seems to have misread or misunderstood the specific statement as well as the more important broader conclusion of the experiment. First, of course the source of GDNF ligand in vivo is the mesenchyme. The explant assay was designed to eliminate this and then to substitute GDNF as provided experimentally. The focus of the experiment was to address the response to GDNF, not the source of GDNF. But more importantly, the experiment revealed a surprising outcome that the reviewer did not appreciate. In *Pax2Cre/Ret* mutants, the Wnt1Cre lineage still expresses Ret, yet does not grow out from the gut explant when provided with GDNF. This shows that the neural crest lineage requires Ret function in placode-derived cells in order to respond to GDNF. In other words, despite expressing Ret, the NC lineage does not harbor a cellautonomous response to GDNF, as we wrote. Because this might be confusing to some readers, we have revised the description of this analysis to hopefully be more clear.

(5) The fact that Ret and Ednrb signaling pathways interact is not a novel finding and has been reported multiple times in Ret and Ednrb mutant mice and cell lines (PMID: 12355085, 12574515 , 27693352, 31818953), potentially through shared transcription factors (PMID:31313802).It would have been more relevant if the authors could show how the specific tyrosine residue (Y 1015) in Ret is phosphorylated in the presence of Ednrb.

The observation that human mutations in RET and EDNRB both cause Hirschsprung disease is decades old, and of course numerous studies in human, mouse, and cells have addressed the relation between the two signaling pathways. We did not mean to imply that we were the first to discover that Ret and Ednrb signaling pathways interact. The reviewer cites a number of papers all from the Chakravarti lab that address this phenomenon; while these are a valuable contribution to the field, there is still more to be learned. The model elaborated in PMID: 31313802, in which Ret and Ednrb are both enmeshed in a common gene regulatory network, does not readily explain why each has a different phenotypic manifestation and doesn’t take into account the importance of the placodal lineage. The main new contributions of our paper are the existence of a new cell lineage that contributes to the ENS, and that the placodal and neural crest lineages utilize Ret and Ednrb signaling differently. The clarification of how these elements are differentially used by the two lineages explains long-segment and short-segment Hirschsprung disease (Ret and Ednrb mutants, respectively) far better than in past studies. The reviewer unfortunately dismisses these insights and seems to feel that a biochemical exploration of one specific component of the signaling interaction (Y1015 phosphorylation) would be more relevant. This should be the basis of future studies and are beyond the scope of the new findings reported in the present paper.

(6) What is the mechanism of the presence of Y1015 phosphorylation in 33% of Ednrb deficient Pax2Cre cells? It appears to me what the authors report as absent phosphorylation in the 67% of cells could be just weak staining or cells missing in prep.

The reviewer, referring to Fig. 7q, presumably meant to say Wnt1Cre rather than Pax2Cre. The reviewer overlooked that we provided an explanation for this observation in our original manuscript. This sentence reads “Because Ednrb is expressed only in a subset of Wnt1Cre-derived enteric progenitor cells (Figure 7 – figure supplement 1), the residual Y1015 phosphorylation observed in Wnt1Cre/Ednrb mutant cells is likely to occur in the Ednrb-negative Wnt1Cre-derived cell population”. The sentence is retained unchanged in the revised manuscript. The explanation is not because of weak staining or problems with tissue preparation.

(7) The references the authors cite regarding the previous discovery of Ret expression in the nucleus are incorrect. The review articles the authors cite do not mention anything about Ret expression in the nucleus. The evidence of nuclear localization of Ret previously comes from overexpression studies in HEK293 cells (PMID: 25795775). Such overexpression studies are fraught with generating noisy data for well-documented reasons. But if this observation is correct, the authors miss a great opportunity to identify what the Ret protein is doing in the nucleus. Is it in direct contact with its known transcription factors like Sox10 and Rarb? This would shed a lot of light on the possible mechanism of Ret LoF observed in Ret mutant mice

The reviewer overlooked that the one of the review articles that we cited (Chen, Hsu, & Hung, 2020) has a dedicated paragraph for RET (section 3.14), which summarizes the work by Barheri-Yarmand et al (PMID: 25795775) which is the very paper noted by the reviewer in the comment above. The reviewer also somewhat misstated the results of the Barheri-Yarmand et al study. By immunostaining, this paper showed nuclear localization of endogenous Ret, albeit a version of Ret with a disease-associated mutation that makes it constitutively active by constitutive autophosphorylation. Nonetheless, this was endogenous Ret. The paper also used overexpression of GFP-tagged RET in HEK293 cells to show that wildtype RET can behave in a similar manner, at least under these circumstances. Our point is simply that Ret (and other receptor tyrosine kinases) can be found in the nucleus in certain biological contexts, and our observations are consistent with this precedent.

The reviewer also suggests a biochemical follow-up analysis related to this observation, which we agree would be of interest. Such an investigation however is beyond the scope of the present study.

(8) The manuscript could benefit from a major rewrite by reorganizing sections to make it easy for the readers to follow the narrative.Many sections about the role of Ret and Ednrb in Wnt1cre-derived ENCDCs can be moved to a supplement. These facts are well-documented and have been proven before.

This was addressed in our response to comment #3 of this reviewer. The figures have been kept as main figures in the revised manuscript to allow side-by-side comparison to parallel analysis of the Pax2Cre lineage.

- The observation that only a handful of Pax2Cre cells at E10.5 express Ret and the observation that conditional Ret null abrogates these cells at E11.5, are not presented together and makes connecting these two facts difficult.

Ret expression at E10.5 and E11.5 are both shown in the same figure (Fig. 2). In the presentation of these results, we first describe in normal development that Ret is expressed differently in E10.5 ENS progenitors between the Pax2Cre and Wnt1Cre lineages. This is additional support for the argument that the two lineages are molecularly distinct. Then comes evaluation of postnatal fates with different markers before we return to embryonic Ret expression. We acknowledge that this can make it difficult to connect these observations. We decided to retain the original organization in order to not lose this important conclusion. However, we have revised the text to hopefully make this connection between the sections more congruent.

**Reviewer #2 (Recommendations For The Authors**):- The labeling of some as "figure supplements" is really hard to follow in the text and confusing to interpret when a main figure or supplemental figure is being referenced, and which one.

We understand this comment, but this is journal style and outside of our control. We have kept the journal format in the revised manuscript.

- The data in Figures 3b-c is well established in the field and somewhat misinterpreted. NOS1 neurons in the mouse ENS and their projections have been well described (Sang and Young, 1996, and other studies). CGRP immunoreactivity would reflect both ENS CGRP-expressing neurons and visceral afferents from DRG.

There of course is a history of analysis of NOS1, CGRP, and other markers in the ENS. The focus of the analysis in Fig. 3 is to demonstrate how the cells that express these markers are impacted by gene manipulation in the Wnt1Cre and Pax2Cre lineages. For the giant migrating contractions that are associated with defecation, ample past electrophysiological studies have established that mechanosensory CGRP+ neurons trigger NOS+ inhibitory neurons (and ACh+ excitatory neurons) of the myenteric plexus to propel colonic contents. Thus, these are the relevant markers to explain the lack of colonic peristalsis in *Ednrb*-deficient mice. To our awareness, our results with NOS1 do not contradict any past study, including the Sang and Young 1996 description. Regarding CGRP, indeed the reviewer is correct that this marker is expressed by both neuronal subtypes. Two arguments support the specific derivation of ENS mechanosensory neurons from the Pax2 lineage. First, the ENS and DRG neurons can be distinguished by the location of their cell bodies and their axon extensions in the gut wall; only the ENS neurons are deficient in Pax2Cre/Ednrb mutants (as documented in Fig. 3). Second, the DRG population is derived from neural crest and is not labeled by Pax2Cre. If this population of CGRP+ neurons had functional relevance to colonic peristalsis, this would not be altered in Pax2Cre/Ednrb mutants. Indeed, the CGRP+ afferent nerve endings of DRG origin in the distal colon are mechanical distension sensors but do not modulate either ENS or autonomic nervous system activity (PMID: 37541195). We believe that our interpretation is correct.

- The evidence in Figure 3 supporting the claim that NOS1 and CGRP-expressing enteric neurons come from distinct lineages is weak. IHC for CGRP is notoriously poor at labeling soma in the ENS. IHC for tdTomato to ensure the detection of low levels of Tomato expression and quantification of observations would strengthen this claim.

CGRP is a vesicular peptide which is stored and transported in vesicles, therefore the antibody against CGRP labels vesicular particles of soma and synaptic vesicles along the axons of those CGRP-producing neurons.

It is not expected to label the entire cytoplasm (or the range of subcellular organelles) as NOS antibody does. We did included quantification of data in Figure 3-figure supplement 1 in the manuscript to support the claim of lineage derivation. As described in the Methods section of the manuscript, we used binary threshold selection for Tomato+ cell count using Fiji-Image J, which detects both TomatoHigh and TomatoLow cells as Tomato+; we feel this is equal to or even superior to IHC for this analysis.

- IHC panels in Figures 3h-o are largely uninterpretable. Most of the signal seems to be non-specific background staining in the mucosa and quantification of mucosal signal in this context does not seem meaningful.

We disagree with the reviewer’s comment. As described in the response above, CGRP+ mechanosensory neurons send their peripheral axon projections to innervate mucosa (sensory epithelial cells), and NOS+ inhibitory motor axons innervate the circular muscle. Thus, panels h-o of Fig. 3 focus on the axonal profile and are not intended to visualize soma, which is why sagittal views are presented instead of flatmount views. All of the controls were performed side-by-side to confirm that the signal is real and interpretable.

Note also that the colon does not have villi so this annotation should be revised.

We appreciate that the reviewer brought this misstatement to our attention. We corrected this error in the revised manuscript.

- Phospho-RET staining in Figure 7 is difficult to discern and interpret with high background. Positive and negative controls would strengthen these data.

Fig. 7 shows phospho Ret-Y1015 staining in lineage-labeled *Wnt1Cre/Ednrb/R26nTnG* mutants. The strength of the signal to noise in the figure is a matter of Ret expression level and the quality of the anti-pY1015 antibody. We are not aware of a meaningful positive control that has been validated in the literature that we could use for comparison. The ideal negative control would be to perform the same analysis in *Wnt1Cre/Ret/R26nTnG* mutants, but because this manipulation eliminates the entire NC cell lineage from the colon, there would be no NC cells in which to visualize background staining in this lineage with this antibody when Ret protein is not present. We note that anti-pY1096 did not show a difference in staining between control and mutant, which supports the interpretation of a specific impact on pY1015. We also point out here, as in the text, that we do not yet have any validation that phosphorylation of Y1015 is functionally important in NC migration to the distal colon. Clearly, more work to address this role and to demonstrate the mechanism of phosphorylation of this specific residue in response to Edn3-Ednrb signaling will be needed.